# Think Big, Teach Small:
# Do Language Models Distil Occam's Razor?

**Gonzalo Jaimovitch-López**[1]  **David Castellano-Falcón**[1]  **Cèsar Ferri**[1]

**José Hernández-Orallo**[1,2]

[1]VRAIN - Universitat Politècnica de València
[2]Leverhulme Centre for the Future of Intelligence - University of Cambridge
`{gonzalojaimovitch,dcastf01}@gmail.com,cferri@dsic.upv.es,jorallo@upv.es`

## Abstract

Large language models have recently shown a remarkable ability for few-shot learning, including patterns of algorithmic nature. However, it is still an open question to determine what kind of patterns these models can capture and how many examples they need in their prompts. We frame this question as a teaching problem with strong priors, and study whether language models can identify simple algorithmic concepts from small witness sets. In particular, we explore how several GPT architectures, program induction systems and humans perform in terms of the complexity of the concept and the number of additional examples, and how much their behaviour differs. This first joint analysis of language models and machine teaching can address key questions for artificial intelligence and machine learning, such as whether some strong priors, and Occam's razor in particular, can be *distilled* from data, making learning from a few examples possible.

## 1  Introduction

An effective and reliable interaction between humans and machines depends on a proper understanding of their priors. The alignment between these priors, in the form of knowledge and other predispositions used as bias, is especially critical for communication and learning (Sun et al., 2020). For instance, after a human utters the *prompt* "on off on off on", another human would expect "off" to be a more likely continuation than "on". In contrast, after the prompt "twinkle twinkle little" we would expect "star" to be more likely. The former depends on an abstract pattern *intrinsic* to the sequence (a simple algorithmic sequence alternating "on" and "off"), while the latter depends on some language pattern *extrinsic* to the sequence (common use in the English language because of a popular song).

Language models are a kind of AI systems extrapolating sequences of symbols, and are very good at exploiting the extrinsic patterns that are borrowed from humans after training over massive natural language corpora. However, it is only recently, with systems such as BERT (Devlin et al., 2018), GPT-2 (Radford et al., 2019) and GPT-3 (Brown et al., 2020), that we see how these models can also discover some intrinsic patterns provided in the prompts. This kind of inference is called 'few-shot learning', even if there is no further training or fine-tuning of the model. Many of these language models are based on transformers and other attention-based architectures (Vaswani et al., 2017), which do not feature recursion or loops, unlike some approaches based on recurrent neural networks. Some particularly well-chosen and optimised prompts have shown remarkable extrapolations (Xu et al., 2020; Izacard and Grave, 2020; Hendrycks et al., 2021), illustrating this new kind of *abstraction on the go*. However, it is unclear how elaborate or algorithmic these abstractions can be. More studies are needed to determine the optimal prompts and example order that make the model capture

35th Conference on Neural Information Processing Systems (NeurIPS 2021).

a range of different patterns (Zhao et al., 2021; Lu et al., 2021; Liu et al., 2021; Webson and Pavlick, 2021; Kirstain et al., 2021). Even when the prompt template is fixed, it is an open question what the minimal set of examples should be for learning a concept, how this minimality is evaluated and what priors pretrained models incarnate for few-shot learning to work.

Precisely, machine *teaching* is the area of artificial intelligence that looks for the optimal examples or curriculum that a teacher should use to make a learner capture a concept (Zhu, 2015). This is of enormous importance today in applications such as digital assistants, where we would like to teach them new concepts or routines with a few examples, in the same way humans teach or communicate with other humans (Cakmak and Thomaz, 2014; Goodman and Frank, 2016; Degen et al., 2020). It is also fundamental for explainable AI, as the dual problem appears: how a machine can teach (explain) a concept to a human in an efficient way (Khan et al., 2011; Basu and Christensen, 2013; Hernández-Orallo and Ferri, 2021). When dealing with rich representation languages (e.g., regular languages or Turing-complete languages), closer or similar to human language in expressive power, it has recently been shown that strong simplicity priors can make effective teaching possible with very few examples: the overall *size* of the example set is usually shorter than the algorithmic expression of the concept itself (Telle et al., 2019). For instance, teaching the concept of what an odd number is becomes *shorter* by using a few examples than by giving the full definition of what an odd number is.

Language models and machine teaching can now meet because of this new few-shot learning ability of language models. We can explore the question of how many examples are needed (or how long they have to be) to *teach* something to a language model. These examples and their template are seen as *teaching prompts*: we want the language model to continue the sequence as if it had captured the pattern that has been expressed in the prompt. From the machine teaching perspective, language models are general systems, as they have not been trained for simplicity or for any particular domain, other than a partial sample of human knowledge expressed as natural language.

With their conjunction, we can now ask key questions for AI such as the relevance of simplicity or acquired priors to make learning from a few examples possible (Chater and Vitányi, 2003). In this paper, we examine how a machine teaching setting inspired by Occam's razor can help us understand how language models 'learn' and choose prompts more efficiently. We use the term 'prior distillation', overloading the notion of *knowledge distillation* within the field of machine learning, as we analyse whether a simplicity prior is acquired *as the result* of training large language models such as GPT-2 or GPT-3. To complete the picture, we also compare them with some other AI systems (e.g., program induction systems) and humans on the same task, which are known to have this prior.

The key contributions of this paper are:

- We analyse language models in the context of machine teaching, using minimal teaching sets for few-shot inference.
- We compare the acquired priors of language models with humans and program induction systems (Gulwani et al., 2015), when facing patterns preferred by a simplicity prior.
- We find that language models such as GPT-3 have similar or better performance than humans. The performance negatively correlates with the complexity of the concept.
- Language models and humans differ in their capability for producing explanations. The length of human explanations correlates with the complexity of the concept.

Having a better understanding of language models and its relation to the way other systems, including humans, make inferences is fundamental research that can make the use of these systems safer and more effective, through more robust prompts to solve a variety of tasks. However, this better understanding can also be used maliciously to cunningly modify prompts that alter the results of some applications built on top of language models. A query may fail when using dates, numbers or other strings that have intrinsic patterns. For instance, GPT-3 usually continues "Mary's father has 5 daughters, Lucy, Anne, Alice, Jane and" correctly, but fails on "Mary's father has 5 daughters, Ba, Baba, Bababa, Babababa and", systematically continued with 'Babababa'.

The rest of the paper follows with related work in language models, the comparison of machine and human learning, as well as Occam's razor and machine teaching. Section 3 contains the experimental design, and introduces the systems we consider for comparison. Section 4 presents and analyses the results. Finally, we close the paper with a discussion of the implications of this work. The supplementary material includes additional experiments, plots, tables and specific analyses.

## 2 Related Work

It is only recently that deep language models are starting to show some minimal capabilities for few-shot learning. Systems such as GPT-3 solve "tasks which require to perform simple on-the-fly computational reasoning, recognize a novel pattern that is unlikely to have occurred in training, or adapt quickly to an unusual task" (Brown et al., 2020). Examples of these tasks are simple arithmetic operations and the induction of symbolic manipulations such as anagrams or reversed words. Hendrycks et al. (2021) compare the performance of a text model with humans on 57 tasks including elementary mathematics, US history, law, and more. Sinha et al. (2019) show that language models obtain poor robustness and generalisation capabilities when compared to graph neural networks that work with symbolic inputs. The experiments also show that humans (with limited time) performed well in simple problems but the performance decreases with complex examples. There are also important limitations of language models on simple generalisation tasks (Bender and Koller, 2020; Marcus and Davis, 2020). On the one hand, recurrent neural networks, such as long-short-term-memory (LSTM) architectures, have shown the capacity of capturing syntax-sensitive generalisations such as long distance number agreement (Lakretz et al., 2019), or language modelling for the game of chess (Toshniwal et al., 2021). The abstract reasoning capacity of transformers and other deep learning architectures has been studied by Hupkes et al. (2020). In sum, as language models based on transformers are not recursive, it is an open debate whether they are just memorising a huge corpus, or they really have the capacity of learning abstract concepts from data.

The comparison between humans and AI systems has usually been performed with systems that are trained for a task, but not for a variety of tasks that are introduced with a few words or examples. For instance, Kühl et al. (2020) present an experiment that compares the performance of several machine learning algorithms and 44 human participants recognising patterns in 3x3 matrices. The results in this case show that humans, as expected, tend to achieve better performance than the AI systems when few examples are provided. This work suggests that many AI systems may not be competitive with humans when limited examples are given for a range of tasks. Further insight can be obtained when we try to look into the extracted models through explanations, and analyse them in the context of concept complexity. For instance, the relation between complexity of concepts and explanations is discussed by Ai et al. (2021).

### 2.1 Occam's Razor

Occam's razor, the a priori preference for simple hypotheses over more complex ones, has pervaded philosophy of science, inductive inference, and human communication, and it is ubiquitous in statistics and machine learning, explicitly —through criteria such as the MML/MDL principles (Wallace and Boulton, 1968; Rissanen, 1983) and the use of regularisation terms— or implicitly —in the optimisation algorithms or parameter-function maps in many methods, including deep learning (Valle-Perez et al., 2019; Shah et al., 2020). Occam's razor is backed by strong theoretical and empirical reasons (Solomonoff, 1964; Dingle et al., 2018), which oppose the idea that all hypotheses should be equally likely a priori, an assumption that would lead to the no free lunch theorems results (Wolpert, 1996), making learning impossible. While simplicity is always relative to a reference language, the invariance theorem (see, e.g., Li and Vitányi, 2008) ensures that the use of different (optimal) languages gives at most a constant difference that only depends on these two languages. While these constants may be very large, some level of agreement in what simplicity is can happen even for very different representations, as we explore in this paper.

A prior for simplicity is hard to understand in large language models, a kind of architecture that is usually overparametrised, compared to the size of the evidence. While the model is not necessarily the most compressed representation of the data, it leads to continuations with low perplexity, which can be used to compress future data, as language models were introduced for communication (Shannon, 1949). It is this interpretation that sheds light on the role of priors, compression and simplicity in human communication and understanding (Chater and Vitányi, 2003; Dowe et al., 2011). It is precisely in the context of few-shot inference, once the model has been trained, that we can reformulate the question as whether these models make inferences based on a preference for simpler hypotheses. We use the term 'distillation', in a somewhat related way to 'knowledge distillation' (Gou et al., 2021), to represent that the simplicity prior has emerged or have been distilled in the trained model, and used as 'learning' bias in few-shot inference. For further discussion about Occam's razor, compression and inference, see Section 15 in the supplementary material.

## 2.2 Machine Teaching and Priors

Machine teaching has analysed learning from a minimal set of examples for different concept classes, with the representational power of the class being a major factor to determine when this is possible. Machine teaching has also been used to understand how humans teach machines, comparing a machine teaching setting with a curriculum learning setting (Khan et al., 2011; Basu and Christensen, 2013). Many of these studies use feature vector concepts instead of rich representations (Turing-complete) such as natural language. Only recently has machine teaching focused on universal languages, and how relevant simplicity priors become when teaching concepts with very few —or very short— examples (Telle et al., 2019; Hernández-Orallo and Telle, 2020; Garcia-Piqueras and Hernández-Orallo, 2021; Hernández-Orallo and Ferri, 2021).

More specifically, machine teaching is often understood as an inverse problem to machine learning (Zhu, 2015), discovering the best set of examples or curricula for a class of concepts and learners. Most of the early research in machine teaching was based on the idea of the *teaching dimension*, the minimum number of examples the learner needs to identify a concept. However, the teaching dimension approach is too narrow. For instance, in the case of teaching concepts from richer languages, it has been demonstrated that even if the teaching dimension is small (low cardinality), the examples might be enormously large. Telle et al. (2019) propose a machine teaching framework that is based on the size of examples instead of the dimension of the witness set. While the teaching dimension is related to algorithmic learning theory, the teaching size is more appropriate for language models and constructive examples. For instance, considering two witness sets of the same teaching dimension, the input-output example $(010101, 1)$ contains more information than $(0, 1)$.

Examples are defined as pairs $(i, o)$ of elements (e.g., binary strings) and concepts as functions mapping inputs to outputs. An example set $S = \{(i_1, o_1), \ldots, (i_k, o_k)\}$ is just a finite set of input-output pairs, used as witness for the teaching process. Considering a representational language $L$ for programs, a learner $\Phi$ can be seen as a function mapping sets of examples to programs. A simplicity-based learner on $L$ just chooses the shortest $L$ program that is compatible with the witness set. As a result, a witness set only needs to distinguish the shortest satisfying program from earlier, shorter ones. Finally, if a teacher wants to teach a concept $c$ it will look for the shortest witness set $S$ (in bit size) such that $\Phi(S)$ returns a program in $L$ that is compatible with $c$. For a more formal definition of the teaching size, the reader is referred to section 14 in the supplementary material.

In this work, we build on the increasing relevance of the alignment of priors between learner and teacher (Goodman and Frank, 2016; Yang and Shafto, 2017; Melo et al., 2018) to investigate how long witness sets have to be for different learners (language models, program induction systems and humans), when devising the minimal teaching example set using simplicity (Occam's razor) as the learner's prior. This is a well-founded controlled experiment setting to analyse the alignment of priors (learning bias) between the source of the concepts (what we call the teacher) and the learner.

## 3 Experimental Setting

The following questions will be explored experimentally (and on average terms):

- Q1 - Abstraction capability: Is a language model such as GPT able to capture algorithmic patterns from very few examples selected with a simplicity prior?
- Q2 - Learning curves: Does accuracy increase from the theoretically minimal witness sets needed for teaching the concept with additional examples?
- Q3 - Characteristic curves: How do language models, program induction approaches and humans behave in terms of the complexity of the patterns?
- Q4 - Similarity: How similar are the errors of language models compared to humans and program induction approaches?
- Q5 - Architecture: Apart from being more capable, which has already been shown before, do larger models have a stronger preference for simple concepts than smaller models?
- Q6 - Explanations: Can we get explanations from GPT for these algorithmic patterns? Is the accuracy of human explanations related to performance, and their size related to complexity?

We now present the language, the machine teaching algorithm and the learners.

## 3.1 The Machine Teaching Setting

The language $L$ we will use to express programs will be the simple but Turing-complete programming language P3, a variation of P'' (Böhm, 1964) with only 7 instructions. We employ P3 as a concept language because it is very different from the representational languages used by the learners we will use (natural language for humans and functional or logical languages for program induction), and especially different from the internal representation of a language model (transformers). This is also a way to see the invariance theorem in action, as all representational languages are different.

P3 programs operate on an input tape and a write-only output tape, with cells containing one of the three characters in the alphabet: $\Sigma = \{$ '0', '1', '.' $\}$. The pointer is moved using the instructions or operators '<' (left) and '>' (right). The instructions '+' and '−' are used to respectively increase or decrease the value of the pointed cell, in the (cyclic) order they are defined in $\Sigma$. The instructions '[' and ']' represent the start and the end of a loop respectively, exiting the loop when the pointer is on '.'. Finally, the instruction 'o' is used to output the value of a pointed cell.

We follow Telle et al. (2019) to generate 'the teaching book': all P3 concepts and the associated witness sets (assuming a simplicity-based learner). The examples are restricted to binary strings, with '.' used as delimiter, padding cells before and after the string. We use a maximum teaching size of 7 (total number of bits in all the inputs and outputs in the witness set). Removing repeated sets and contradictory examples, the example set space is hence comprised of 17,252 example sets.

## 3.2 Choice of Concepts and Examples

We extract eight different concepts $c_1, \ldots, c_8$, each $c_i$ being randomly selected from all programs in the teaching book with $i$ instructions. We bin the concepts into four levels: the programs with 1-2 operators (very low complexity, VL-C), those with 3-4 operators (low complexity, L-C), those with 5-6 operators (high complexity, H-C) and those with 7-8 operators (very high complexity, VH-C).

The minimal witness sets for these concepts are all between 1 and 3 examples. This will be the first batch (the witness set, WS) used as training set. This would lead to perfect identification if the learner were using the same simplicity prior over the same language. But, as the learner's prior might differ from the teacher's, we also want to analyse how learning improves as more examples are provided. Therefore, we will use 5 additional examples in two different batches, a first batch with 2 extra examples (additional set one, AS I) and a second batch with 3 extra examples (additional set two, AS II). These two additional sets cannot follow the machine teaching protocol (as they are redundant for perfect alignment). Instead of measuring their size, but to keep it in a similar distribution as the rest, we extract *new* examples from the example sets for the same program in the teaching book. For the test examples, we generate 5 new examples per concept, using random sampling for the inputs $i$ (avoiding repeated strings in WS, AS I and AS II) with probability $2^{-(l(i)+1)}$. In total, we have $8 \times 3 \times 5 = 120$ predictions. This number of concepts, batches and test examples aims to find a trade-off for a minimally robust evaluation and the limitations of conducting the experiments with humans.

Table 1: Experimental setting for the learning of concepts $c_1$ and $c_4$.

| Id | P3 | Description (not used) | WS | AS I | AS II | Test |
|----|----|------------------------|----|------|-------|------|
| $c_1$ | o | *Print the first character of a string* | {('0','0')} | {('111001','1'), ('110101','1')} | {('100110','1'), ('01010','0'), ('111100','1')} | {('00000','0'), ('11100','1'), ('00111','0'), ('11010','1'), ('0010', '0')} |
| $c_4$ | o+oo | *If input is '', print '00'; else if input starts with '0', print '011'; else print '1'* | {('0','011')} | {('10','1'), ('001','011')} | {('00','011'), ('0001','011'), ('000','011')} | {('01011','011'), ('0101','011'), ('0010','011'), ('100','1'), ('1','1')} |

Table 1 shows two concepts, $c_1$ and $c_4$, their code in $P3$, an intuitive description, and the several sets (WS, AS I, AS II and Test). The Tables 1, 2, 3 and 4 in the supplementary material contain all used concepts. With all this, the proposed teaching protocol for each 'concept lesson' is divided into three phases: (1) we ask to predict the test examples after only seeing WS, (2) we ask to predict the same 5 test examples after seeing WS + AS I and (3) we ask to predict the same 5 test examples after seeing WS + AS I + AS II.

### 3.3 Learners

The following learner families have been considered.

**Humans:** 30 volunteering (unpaid) human participants with ages between 18 and 54. Apart from the answers to each test example, humans had to write an explanation in natural language for each of the 8 concepts after the third phase. All data collected in the experiments were anonymised, informing the participants of the data treatment and their rights, following data protection laws. The form can be found in the GitHub repository (link at the end of the next section).

**MagicHaskeller** (**MH**): A general-purpose inductive functional programming system based on systematic exhaustive search (Katayama, 2011). It infers programs expressed in Haskell from input-output examples. A strong simplicity prior and a library of primitives allow MH to solve many problems using only one example. When MH returns more than one hypothesis, we take the first one.

**Louise**: An inductive logic programming system based on Top program construction (Patsantzis and Muggleton, 2021). It learns Prolog programs from finite examples, with the capability of learning recursive programs in polynomial time. Two different approaches are considered: **Louise-ens**, where the most frequent result returned by the totality of the Top program will be used as the final answer, with ties are broken randomly, and **Louise-exp**, where the expected score is calculated considering all the possible solutions returned by the Top program.

**GPT-2**: A transformer-based language model trained on a dataset of 8 million web pages (Radford et al., 2019). The model returns the next predicted word or token considering the previous tokens as the given context. We use the 774M-parameter version of GPT-2. The default hyperparameters of the model are preserved, with the exception of the 'length' of the output returned, which is modified to return 3 tokens. By doing so, the experiment execution performance is improved, without asking GPT-2 for more information than what we require for our analysis. Deterministic results can be obtained by setting the 'top-k' parameter to 1, but we can get better performance if we use several continuations. Accordingly we will use the default 'top-k' value, which is 0. With this, two different approaches will be considered by running the model extrapolations $n=40$ times for the same query: **GPT-2-ens**, where the most frequent result (out of the $n$ results obtained for the same query) will be used as the final answer, with ties are broken randomly (this approach can be understood as a majority ensemble setting), and **GPT-2-exp**, where the $n$ results for the same query are averaged (this approach can be understood as the expected value for GPT-2).

**GPT-3**: Another series from the GPT family (Brown et al., 2020). We used different available models, with an increasing number of parameters (Ada, Babbage, Curie and Da Vinci), as recent work (Bender et al., 2021) demonstrates that larger models outperform smaller ones. We want to show whether this trend holds for abstract concepts too, with Occam's razor emerging more strongly for larger models. We will employ acronyms GPT-3A, GPT-3B, GPT-3C, GPT-3D correspondingly. We performed the experiment considering two different settings: the hyperparameter temperature set to 0 and to 1. When the temperature is set to 1, the results are non-deterministic, and the same approach used in GPT-2-ens is applied. We will use T0 or T1 as shortcuts depending on the value of this parameter. For instance, GPT-3 Da Vinci with temperature 0 will be denoted by GPT-3D-T0. Finally, we also asked GPT-3 to provide an explanation of the concepts, just after each batch ($8 \times 3$ explanations).

We explored very different kinds of prompts for both GPT-2 and GPT-3. A complete account of all the prompts we tried can be found in Table 9 in the supplementary material. Table 2 here shows the prompts that gave the best results, consequently being used in the experiments.

Table 2: Final prompts used in the experiments with GPT-2 and GPT-3.

| System | Prompt | Observations |
|---|---|---|
| GPT-2 | Input1: 0, Output1: 0; Input2: 00000, Output2: | The numbering aims to logically connect and distinguish the different input-output pairs. |
| GPT-3 | Input: 0
Output: 0
Input: 00000
Output: | The separator is the line break with no numbering, following the prompt style guidelines from OpenAI API. |

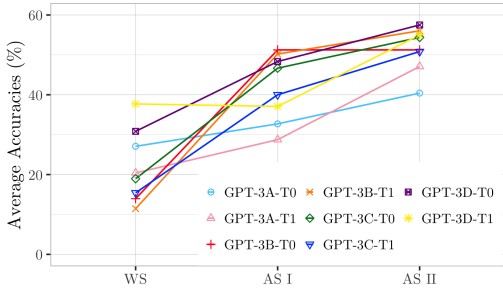

(a) Mean GPT-3 accuracy by teaching batch.

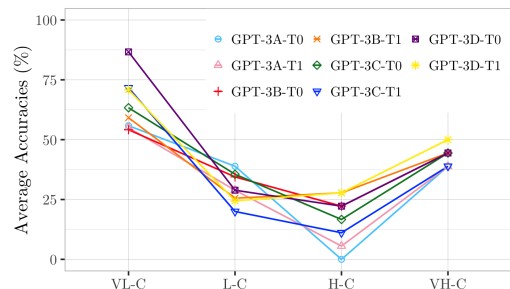

(b) Mean GPT-3 accuracy by concept complexity level.

Figure 1: Results for eight GPT-3 configurations. Figure (a) shows the learning curves: average accuracies when aggregated for the eight concepts (1-8 instructions) and split by the three teaching set batches. WS refers to the first (optimal) teaching set, AS I refers to these plus two additional examples and AS II refers to all these plus the final three additional examples. Figure (b) shows the characteristic curves: the average accuracies when aggregated for the three teaching set batches and split by the four concept complexity levels. VL-C, L-C, H-C and VH-C refer to programs with 1-2, 3-4, 5-6 and 7-8 instructions, respectively. A less crowded version of this figure, ablated by temperature and model size can be found in Figures 1 and 2 in the supplementary material.

## 4  Results

We first report the results of the main experimental setting that compares humans, language models and other systems, restricted to 8 concepts (test examples with the empty string as output not included in the means). More extensive results with a larger number of concepts at the end of section.

### 4.1  Comparison against Humans using 8 Concepts

We take the results of each of the learners in the previous section. For humans, we aggregate the results into a single value per concept, batch and test example, by calculating the average of the 30 human subjects. Performance is represented using learning curves and characteristic curves. A learning curve depicts the variation of accuracy according to the number of examples, which in our case is grouped into the batches WS, AS I and AS II. Characteristic curves depict the variation of accuracy in terms of the complexity level of the concepts: VL-C, L-C, H-C and VH-C. Accuracies are always reported as a percentage of the test examples that are correctly predicted.

We start with the analysis for the eight GPT-3 parameter configurations. The learning curves in Figure 1a are clearly increasing in general. The first observation is that the systems with the temperature set to 0 obtain better performance than those with the temperature set to 1. There is no clear evidence that larger models in terms of the number of parameters present a better performance for this kind of patterns, which are solely based on simplicity and not other priors. Regarding the characteristic curves in Figure 1b, the GPT-3 systems show a decreasing trend in performance as the complexity increases. Once again, there is no clear pattern in terms of the number of the parameters. Nevertheless, GPT-3D-T0 is the best method, and both GPT-3D versions (the largest models) show their advantage in the very few shot case (WS) and low complexity (VL-C). In other words, more parameters seem to make a difference for easier concepts with fewer examples.

Figure 2a shows the learning curves for humans, MH, the two variations for Louise and a selection of GPT methods. GPT-3D-T0 turns out to be the system with the highest average performance for every teaching batch. Louise's variants follow, with a similar performance between them. Humans and GPT-2-ens go next, with similar general performance in the experiment. GPT-2-exp remains slightly under the average performance of the ensemble (majority) approach. Finally, MH is the poorest system, especially for WS, the case with very few examples. For all systems, the accuracy increases as additional examples are provided (AS I and AS II).

Figure 2b shows the characteristic curves. We see curves cross more often, but overall the order of the learners in terms of performance is similar to the learning curves. The evolution of performance in terms of complexity is generally decreasing, with the best results clearly being for concepts of

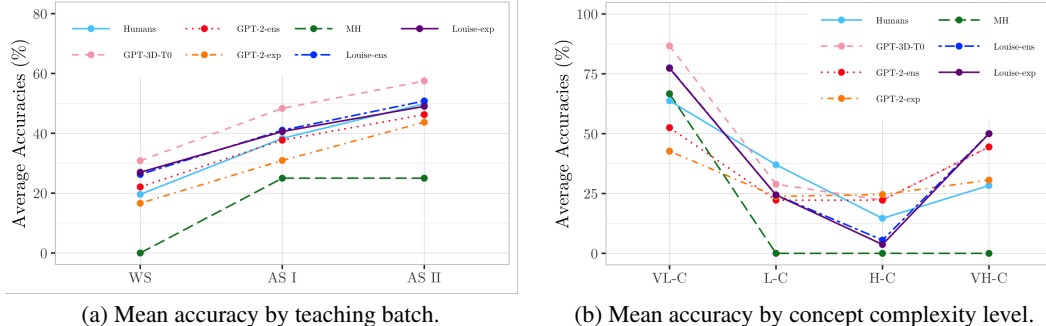

(a) Mean accuracy by teaching batch.

(b) Mean accuracy by concept complexity level.

Figure 2: Figure (a) presents the average accuracies for seven learning agents shown for the three teaching set batches. Interpretation as in Figure 1a. Figure (b) presents the average accuracies for seven learning agents shown for the four complexity levels. Interpretation as in Figure 1b.

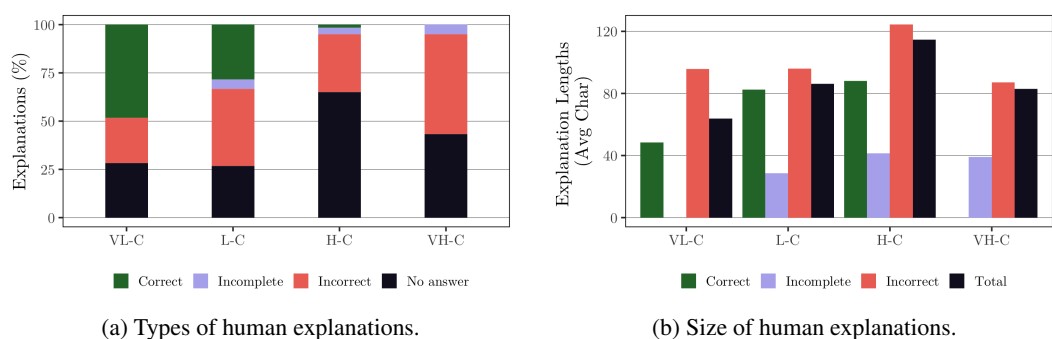

(a) Types of human explanations.

(b) Size of human explanations.

Figure 3: Figure (a) shows a breakout of the quality of human explanations after batch AS II. Correct answers are those that explain the whole concept correctly, incomplete answers explain part of the concept correctly and incorrect answers explain any part of the concept incorrectly. Figure (b) shows the average lengths of the different explanations provided by the human participants after batch AS II.

very low complexity (VL-C). Looking into the results in more detail (see supplementary material), we find different reasons for the small increase in the last group (VH-C), but it is mostly explained because the chosen test examples have short outputs for these two concepts. This increase for VH-C disappears when experiments are performed with a large number of concepts, as we discuss below.

Explanations for the given predictions may be a good way of better understanding whether alignment has taken place, beyond accuracy. When looking for explanations, we tried many kinds of prompts for the GPT systems, as discussed in the supplementary material. However, we did not manage to get good explanations. For Louise, despite being interpretable, the solutions usually had too many clauses to serve as explanations. Only for MH the solution programs are easily interpretable by humans, assuming some knowledge of functional programming, but MH was seldom correct. Because of all this, we will pay more attention to the explanations given by humans in the questionnaire.

We consider a correct explanation if, after inspection by us, the whole behaviour of the concept is well explained and is totally correct. We consider it incomplete if it is a partial account of the concept. We consider it incorrect if any part of it does not correspond to the concept. Note that the explanation might be wrong and the inferences correct, and the other way round. As observed in Figure 3a, the percentage of correct natural language explanations decreases as the complexity of the concepts increases. On the other hand, Figure 3b shows how the average size of the correct explanations increases with the complexity of the concepts (size is measured as the number of characters in the natural language explanation given by humans). Remarkably, the average size of the different types of explanations present the opposite behaviour to performance considering the concept complexity bins in Figure 2b. Even for incorrect explanations, the perceived complexity in terms of the size of the explanation inversely corresponds to the accuracy results.

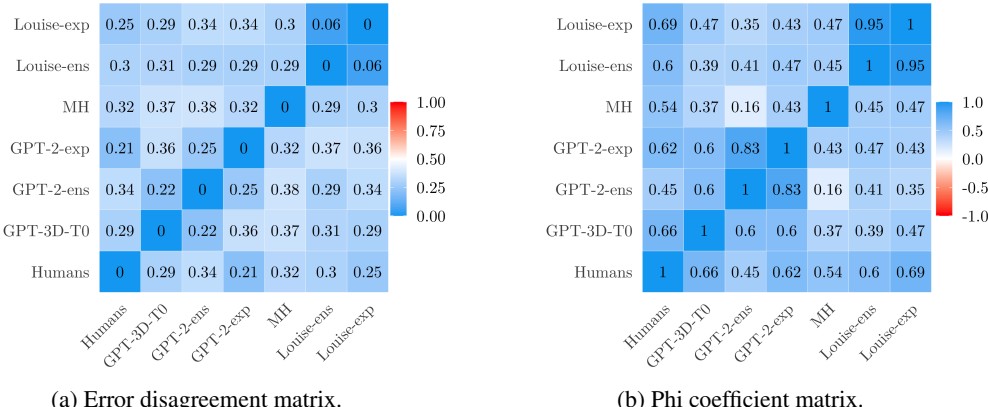

(a) Error disagreement matrix.                (b) Phi coefficient matrix.

Figure 4: (a) Error disagreement matrix for performance and (b) Phi coefficient matrix for the behaviour of seven learning agents.

In the previous plots, we have compared learners by batches of examples or complexity of the concepts. However, if we want to have a more fine-grained metric of how much similar the systems are, we can analyse if they succeed on the same examples. This is what we see in Figure 4, showing the error disagreement (proportion of cases where one method is right and the other is wrong, or vice versa) (Figure 4a) and the Phi coefficient (Figure 4b) taking all the examples individually. The matrix for error disagreement strongly depends on performance, so the results for Phi coefficient (Pearson correlation) are more informative. We see that approaches of the same family usually have high correlations. When compared with humans, GPT-3D-T0 and Louise-exp are the most similar ones.

## 4.2 Results for Language Models for Larger Concept Samples

The supplementary material includes more details of the comparison between humans, language models and the program induction systems. While comparison against humans is limited by the number of concepts that humans could attempt in a single form, we do not have these limitations when using language models. In particular, the supplementary material includes concepts with and without loops. We see that GPT-3 is better for concepts without loops, especially when sufficient examples are given in phase three (WS + AS I + AS II).

Here we just report the results for much larger samples. Table 3 includes the results considering 300 concepts (shown also in the supplementary material with other parameter sizes in Tables 15, 16, 17 and 18). What we see is a much more gradual increase of performance as more information is given through the three batches, and also a decrease of performance as the concepts are more complex.

Table 3: Average GPT-3D-T0 accuracy for 300 concepts.

| GPT-3D-T0 | Batch | VL-C (%) | L-C (%) | H-C (%) | VH-C (%) | Row Mean |
|---|---|---|---|---|---|---|
| | WS | 37.00 | 20.05 | 6.71 | 10.02 | 18.44 |
| | AS (I) | 47.00 | 38.56 | 18.76 | 19.16 | 30.87 |
| | AS (II) | 75.00 | 46.70 | 29.55 | 22.35 | 43.40 |
| Column Mean | | 53.00 | 35.10 | 18.34 | 17.18 | |

The supplementary material also includes results for the majority and random baselines (discarding spurious effects) and experiments not using machine teaching. All the code and experimental data can be found at `https://github.com/gonzalojaimovitch/think-big-teach-small`.

# 5 Discussion

We can now answer the questions we formulated in Section 3. We see that only for easy concepts (VL-C), the simplicity prior leads to example sets that humans, Louise and some GPT models can identify consistently (Q1, Q3). For more complex concepts the behaviour is more erratic, and may depend on spurious factors, as the explanations are incorrect for those very hard cases having some accuracy. The learning curves are increasing for all systems, as more examples are given (Q2). The minimal set is rarely sufficient, which means that no system prior is perfectly aligned with the simplicity prior that derives from P3 program size. This is expected as the internal representations of these systems are different. Both humans and GPT have priors to capture both extrinsic and intrinsic patterns, as language models have been trained using natural language texts created by humans. This seems consistent with large language models having a comparable performance to humans. However, this would not explain cases in which GPT-3 outperforms humans, unless (1) our population behaves differently (worse) than the population behind the training corpora used for learning them, and/or (2) GPT-3 has generalised some very abstract latent concepts that are applicable for simple intrinsic patterns, where GPT-3 seems to excel (VL-C). Finally, MH is the system that behaves most like an ideal learner in our setting, but with very limited capability. The difference in the performance between MH and Louise might be due to the differences in their background knowledge.

Considering both matrices in Figure 4, GPT-3D-T0 shows a similar pattern of behaviour (correct/incorrect results) with respect to humans in comparison with the rest of the analysed systems (Q4). This means the extra performance is not explained by being more deviated from human priors, but perhaps because of higher robustness. When we look at architecture size, we do not see a monotonic relation between results and number of parameters (Q5), but GPT-3D, for the two variants that we have studied, achieves the best results for simple concepts. This may suggest that *larger models can be better at capturing patterns that depend on simplicity priors, distilling some kind of Occam's razor for few-shot inference*. However, some other experiments in the supplementary material, especially those concepts including loops, suggest that larger models in the future may not be able to solve complex algorithmic concepts, with this distillation being limited to sequential (if-then-else) algorithmic patterns. New paradigms beyond transformers may be needed in the future.

Explanations should give us more insight about what these systems are doing. Unfortunately, we were not able to find a working prompt so that GPT-3 could generate good explanations for the learnt concepts (Q6). Regarding the explanations provided by humans, the results suggest that the higher the complexity, the higher the size of the correct explanations to cover the learnt concepts. Furthermore, as complexity increases, the percentage of correct explanations decreases.

There are some limitations in this work. First, while we use MH as a system that would only be dominated by the simplicity priors, it is not very capable, and it is only useful to disentangle the behaviour for very easy concepts. Louise adds robustness to this analysis but it has more dependency on background knowledge. More inductive programming systems could help in this analysis. Second, the use of humans for comparison limits the number of concepts and test examples that we can do for a reasonable questionnaire. This has been partially mitigated by more concepts being explored for GPT-3 (section 4.2 and supplementary material), but the analysis of different samples according to other characteristics (not only loops but the frequencies of symbols or the lengths of the strings) could further characterise the performance. Third, the machine teaching setting we have used assumes perfect alignment and perfect identification, and this does not help in the understanding of how to provide further examples (AS I and AS II). Similarly, for compatibility with the original teaching book, in case of equal $l$, ties were broken lexicographically, which adds more dependence on P3. Fortunately, the machine teaching setting can be extended probabilistically for the teaching dimension (Hernández-Orallo and Telle, 2020) (with more evidence increasing the confidence of a correct identification), and a similar extension for the teaching size could be used for this analysis.

The main take-away of this paper is a new approach, based on machine teaching, to analyse the power of language models under few-shot inference. Because of the complexity of transformer architectures, the diversity of the training data used for language models, and their use as pre-trained models through very different kinds of prompts, we require empirical yet well-founded ways to analyse how much evidence is needed for these systems to make the right extrapolations. This analysis boils down to determining the priors that pretrained models have distilled and the bias they represent. Overall, machine teaching can bring very important insights about what language models can do, what their acquired priors are, and the number —or the size— of examples they need.

## Acknowledgments

We thank OpenAI for giving us early access to GPT-3 through their API. We thank support by the EU (FEDER) and Spanish grant RTI2018-094403-B-C32 funded by MCIN/AEI/10.13039/501100011033 and by "ERDF A way of making Europe", Generalitat Valenciana under grant PROMETEO/2019/098 and INNEST/2021/317 (Project cofunded by the European Union with the "Programa Operativo del Fondo Europeo de Desarrollo Regional (FEDER) de la Comunitat Valenciana 2014-2020"), EU's Horizon 2020 research and innovation programme under grant agreement No. 952215 (TAILOR), the FLI under grant RFP2-152, and US DARPA HR00112120007 (RECoG-AI). We thank the anonymous reviewers for their comments and interaction during the discussion process. We are grateful to Dan Hendricks, Fernando Martínez-Plumed and Richard Evans for useful discussions, Carlos Monserrat for some assistance with the plots, Lidia Contreras-Ochando for her help with MagicHaskeller and María José Ramírez-Quintana for their help with questions about visualisation, inductive programming and Louise. We also thank Stephen Muggleton and Stassa Patsantzis for their help with Louise. Finally, we thank the participants who completed the questionnaire.

## Funding Transparency Statement

All sources of funding have been included in the acknowledgments. None of the authors have financial relationships with entities that could potentially be perceived to influence the objectivity of the paper.

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
