# Think Big, Teach Small:
# Do Language Models Distil Occam's Razor?

# SUPPLEMENTARY MATERIAL

**Gonzalo Jaimovitch-López**[1]
gonzalojaimovitch@gmail.com

**David Castellano-Falcón**[1]
dcastf01@gmail.com

**Cèsar Ferri**[1]
cferri@dsic.upv.es

**José Hernández-Orallo**[1,2]
jorallo@upv.es

[1]VRAIN - Universitat Politècnica de València
[2]Leverhulme Centre for the Future of Intelligence - University of Cambridge

## Introduction

In this document, we present additional information providing a deeper understanding of certain aspects of the paper. We include some other experiments not described in the paper. Specifically, we present:

1. **Concepts and Examples.** The concepts in P3 and the complete teaching and test sets used in the experiments (Table 1)

2. **Concepts in the Form of Decision Rules.** The P3 concepts expressed in the form of decision rules for better understanding (Tables 2, 3 and 4).

3. **Ablation by Temperature and Model Size.** We split Fig. 1 in the paper by temperature and model size by showing T0 in Fig. 1 and T1 in Fig. 2.

4. **Complete Experimental Results.** The experimental results for all learners considering the different teaching batches and concept complexities (Tables 5, 6, 7 and 8).

5. **Explored Prompts.** Examples of the prompts used in the experiments with GPT-2 and GPT-3 (Tables 9 and 10).

6. **Detailed Results with Systems with Variability**. These report the performance and error bars for those systems that present variability (GPT-3A-T1, GPT-3B-T1, GPT-3C-T1, GPT-3D-T1, GPT-2-exp) including humans as well because of populational variability (Figures 9, 10, 7, 8, 5, 6, 3, 4, 11, 12, 13 and 14).

7. **Additional Experiments: Concepts with Loops.** The results obtained by GPT-3D-T0 in the learning of four different algorithmic concepts using loops (Table 11).

8. **Additional Experiments: Concepts without Loops.** The results obtained by GPT-3D-T0 for three new concepts with H-C and VH-C complexities (Tables 12 and 13) (Figures 15 and 16).

9. **Additional Experiments: Many More Concepts and Baselines.** This first reports how many concepts we can have for each complexity group (Table 14). This determines the experiments with extra concepts (Tables 15, 16, 17 and 18 with 300 concepts). Tables 19 and 20 show the majority and random baselines.

35th Conference on Neural Information Processing Systems (NeurIPS 2021).

10. **Additional Experiments: Not Using Machine Teaching.** This section includes results with a different way of generating the training examples, not based on machine teaching. Tables 21 and 22 compare the generation based on machine teaching with an alternative generation protocol.

11. **Explanations produced by GPT-3.** Here we include a sample of the explanations from the learned concepts produced by GPT-3 (Tables 23, 24, 25 and 26).

12. **Software.** Availability and licenses of the software used in the experiments (Table 27).

13. **Hardware specifications.** Specification of the hardware used for the experiments (Table 28).

14. **Formal Definition of Teaching Size and the Teaching Book.** This includes a formal account of teaching size based on (Telle et al., 2019).

15. **Distilling Simplicity, and Occam's Razor for Few-shot Inference.** An extension of the coverage and discussion of related work about simplicity in deep learning (and transformers) and the contingency of the hypothesis we explore in this paper.

16. **Program Length Ties and Lexicographic Preference.** This section discusses the effect of choosing lexicographic preference for programs and strings (witness sets).

The data and code can be found in https://github.com/gonzalojaimovitch/think-big-teach-small.

# 1 Concepts and Examples

We include the complete set of concepts with their teaching (witness set WS, additional set AS I and additional set AS II) and test examples.

Table 1: Complete set of concepts with their teaching (WS, AS I and AS II) and test examples.

| | | | Experimental Setting | | |
|---|---|---|---|---|---|
| Id | P3 Program | Witness Set | Additional Set I | Additional Set II | Test Examples |
| C1 | o | {('0','0')} | {('111001','1'), ('110101','1')} | {('100110','1'), ('01010','0'), ('111100','1')} | {('00000','0'), ('11100','1'), ('00111','0'), ('11010','1'), ('0010', '0')} |
| C2 | >o | {('10','0')} | {('01000','1'), ('01010','1')} | {('1011','0'), ('00','0'), ('001','0')} | {('01011','1'), ('0101','1'), ('0010','0'), ('100','0'), ('1','')} |
| C3 | >+o | {('','0'), ('01','')} | {('010100',''), ('10101','1')} | {('010',''), ('100','1'), ('011101','')} | {('00010','1'), ('110',''), ('00111','1'), ('11000',''), ('101','1')} |
| C4 | o+oo | {('0','011')} | {('10','1'), ('001','011')} | {('00','011'), ('0001','011'), ('000','011')} | {('01011','011'), ('0101','011'), ('0010','011'), ('100','1'), ('1','1')} |
| C5 | >>>-o | {('','1'), ('1110','')} | {('10','1'), ('111001','')} | {('11','1'), ('11100',''), ('0001','0')} | {('01011','0'), ('110','1'), ('0010',''), ('101','1'), ('1000','')} |
| C6 | >-[o<] | {('0','10'), ('00','')} | {('11','01'), ('10','')} | {('101',''), ('','1'), ('000','')} | {('01','00'), ('0000',''), ('00011',''), ('0011',''), ('1000','')} |
| C7 | -[-<]>o | {('','0'), ('0',''), ('00','0')} | {('0001','0'), ('01','1')} | {('0101','1'), ('0010','0'), ('0110','1')} | {('01011','1'), ('0000','0'), ('00000','0'), ('100',''), ('1000','')} |
| C8 | +[>+o<+] | {('','01'), ('01',''), ('1','')} | {('11',''), ('011','')} | {('10',''), ('0','0'), ('0100','')} | {('10101',''), ('11101',''), ('00000','1'), ('0011','1'), ('1111','')} |

# 2 Concepts in the Form of Decision Rules

The concepts used in the experiment were originally in P3. Here, for the sake of comprehensibility, we show the concepts expressed as decision rules in Python.

Table 2: Description of P3 concepts 1-4 using Python decision rules.

| Id | P3 Programs | Decision Rules |
|----|-------------|----------------|
| 1 | o | ```python
if input == '':
    print('')
else:
    print(input[0])
``` |
| 2 | >o | ```python
if input == '' or
   input == '0' or
   input == '1':
    print('')
else:
    print(input[1])
``` |
| 3 | >+o | ```python
if input == '' or
   input == '0' or
   input == '1':
    print('0')
elif input[1] == '0':
    print('1')
else:
    print('')
``` |
| 4 | o+oo | ```python
if input == '':
    print('00')
elif input[0] == '0':
    print('011')
else:
    print('1')
``` |

Table 3: Description of P3 concepts 5-7 using Python decision rules.

| Id | P3 Programs | Decision Rules |
|----|-------------|----------------|
| 5 | >>>-o | |

```
if input == '' or
   len(input) <= 3:
    print('1')
elif input[3] == '0':
    print('')
else:
    print('0')
```

| 6 | >-[o<] | |

```
if input == '':
    print('1')
elif input == '0':
    print('10')
elif input == '1':
    print('11')
elif input[1] == '0':
    print('')
elif input[:2] == '01':
    print('00')
else:
    print('01')
```

| 7 | -[-<]>o | |

```
if input == '':
    print('0')
elif input == '0' or
     input[:1] == '1':
        print('')
else:
    print(input[1])
```

Table 4: Description of P3 concept 8 using Python decision rules.

| Id | P3 Programs | Decision Rules |
|----|-------------|----------------|
| 8  | +[>+o<+]    | |

```python
if input == '':
    print('01')
elif input == '0':
    print('0')
elif input[:1] == '1' or
    input[:2] == '01':
    print('')
else:
    print('1')
```

# 3 Ablation by Temperature and Model Size

Here we include the same results as Fig. 1 in the main paper but separated by temperature and model size, showing T0 in Fig. 1 and T1 in Fig. 2.

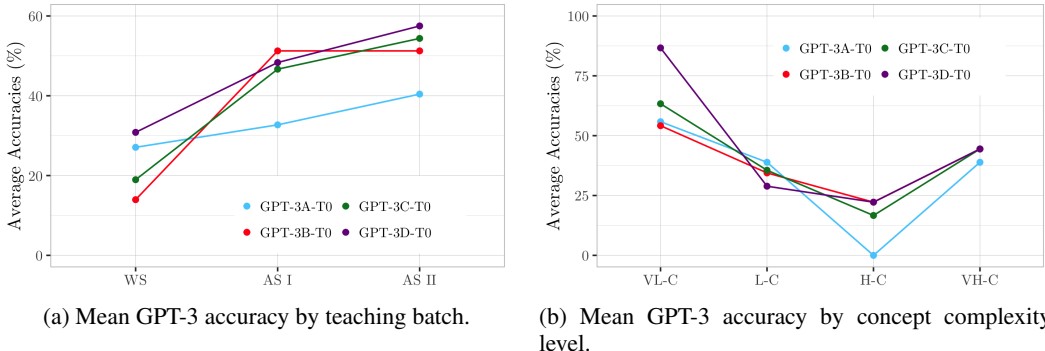

(a) Mean GPT-3 accuracy by teaching batch.

(b) Mean GPT-3 accuracy by concept complexity level.

Figure 1: Same as Fig. 1 in the main paper but only for T0.

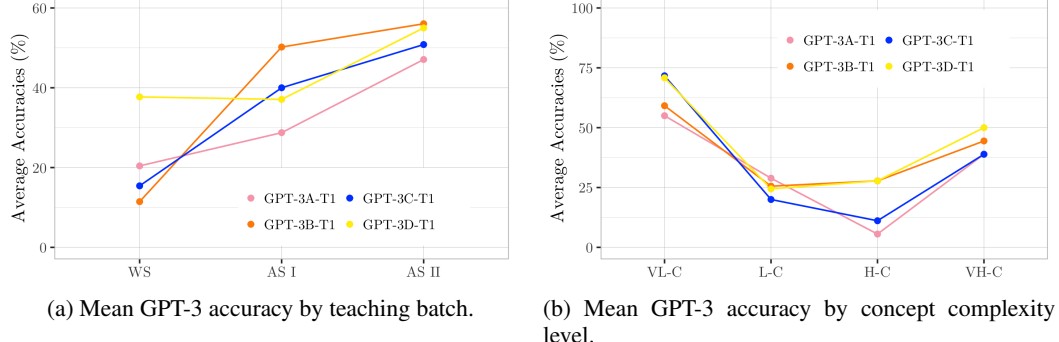

(a) Mean GPT-3 accuracy by teaching batch.

(b) Mean GPT-3 accuracy by concept complexity level.

Figure 2: Same as Fig. 1 in the main paper but only for T1.

# 4 Complete Experimental Results

Here we include the experiment results for all learners considering the teaching batches and concept complexities.[1] Note that Table 5 and Table 7 include *all* the results, while Table 6 and Table 8 showcase the results when the test examples whose output is the empty string are excluded, as we do in the paper. We discard test examples with the empty string as output because humans usually leave it blank when they do not know the answer, and the language model's outputs are postprocessed in such a way that the empty output can simply capture meaningless outputs from GPT.

Table 5: Average accuracy obtained by all learners with concepts binned by complexity (from very low to very high) and by the three batches.

| Learner | Batch | VL-C (%) | L-C (%) | H-C (%) | VH-C (%) |
|---|---|---|---|---|---|
| Humans | WS | 32.00 | 23.00 | 44.00 | 44.65 |
| | AS (I) | 63.35 | 42.00 | 43.65 | 48.65 |
| | AS (II) | 81.00 | 48.65 | 45.00 | 51.65 |
| MH | WS | 10.00 | 20.00 | 20.00 | 0.00 |
| | AS (I) | 100.00 | 0.00 | 0.00 | 0.00 |
| | AS (II) | 100.00 | 0.00 | 0.00 | 0.00 |
| Louise-ens | WS | 50.00 | 0.00 | 10.00 | 60.00 |
| | AS (I) | 70.00 | 30.00 | 0.00 | 60.00 |
| | AS (II) | 100.00 | 30.00 | 10.00 | 60.00 |
| Louise-exp | WS | 52.32 | 0.00 | 10.00 | 45.00 |
| | AS (I) | 71.29 | 30.00 | 0.00 | 60.00 |
| | AS (II) | 95.83 | 30.00 | 6.66 | 60.00 |
| GPT-2-ens | WS | 40.00 | 30.00 | 40.00 | 40.00 |
| | AS (I) | 50.00 | 10.00 | 60.00 | 60.00 |
| | AS (II) | 50.00 | 40.00 | 70.00 | 60.00 |
| GPT-2-exp | WS | 17.00 | 12.25 | 26.00 | 24.25 |
| | AS (I) | 48.75 | 19.25 | 32.00 | 38.00 |
| | AS (II) | 49.25 | 40.75 | 42.50 | 43.75 |
| GPT-3D-T0 | WS | 70.00 | 30.00 | 60.00 | 50.00 |
| | AS (I) | 70.00 | 40.00 | 50.00 | 60.00 |
| | AS (II) | 90.00 | 40.00 | 60.00 | 60.00 |

---

[1]The complete code for the experiments can be found at https://github.com/gonzalojaimovitch/think-big-teach-small

Table 6: Average accuracy obtained by all learners in the teaching task with concepts binned by complexity (from very low to very high) and batches, not including the test examples where the output is the empty string.

| Learner | Batch | VL-C (%) | L-C (%) | H-C (%) | VH-C (%) |
|---|---|---|---|---|---|
| Humans | WS | 38.17 | 13.00 | 7.78 | 19.44 |
| | AS (I) | 67.92 | 43.89 | 12.78 | 28.61 |
| | AS (II) | 85.17 | 54.11 | 23.33 | 36.94 |
| MH | WS | 0.00 | 0.00 | 0.00 | 0.00 |
| | AS (I) | 100.00 | 0.00 | 0.00 | 0.00 |
| | AS (II) | 100.00 | 0.00 | 0.00 | 0.00 |
| Louise-ens | WS | 55.00 | 0.00 | 0.00 | 50.00 |
| | AS (I) | 77.50 | 36.60 | 0.00 | 50.00 |
| | AS (II) | 100.00 | 36.60 | 16.60 | 50.00 |
| Louise-exp | WS | 57.87 | 0.00 | 0.00 | 50.00 |
| | AS (I) | 75.74 | 36.60 | 0.00 | 50.00 |
| | AS (II) | 98.33 | 36.60 | 11.10 | 50.00 |
| GPT-2-ens | WS | 45.00 | 10.00 | 0.00 | 33.33 |
| | AS (I) | 57.50 | 10.00 | 33.33 | 50.00 |
| | AS (II) | 55.00 | 46.67 | 33.33 | 50.00 |
| GPT-2-exp | WS | 18.63 | 6.83 | 18.33 | 22.71 |
| | AS (I) | 54.63 | 17.00 | 24.58 | 27.50 |
| | AS (II) | 54.81 | 47.50 | 30.83 | 41.62 |
| GPT-3D-T0 | WS | 80.00 | 10.00 | 0.00 | 33.33 |
| | AS (I) | 80.00 | 30.00 | 33.33 | 50.00 |
| | AS (II) | 100.00 | 46.67 | 33.33 | 50.00 |

Table 7: Average accuracy obtained by the eight different GPT-3 configurations in the teaching task with concepts binned by complexity (from very low to very high) and by the three batches. The system GPT-3D-T0-S is a variation of the GPT-3D-T0 configuration, with a prompt modification separating the string characters using a specific separator, in order to address a possible byte-pair encoding implicitly implemented in the system. The system GPT-3D-T0-A is a variation of the GPT-3D-T0 configuration, with a prompt modification where the character "0" was replaced with the string "Apple", the character "1" was replaced with the string "Banana", and the character "0" was replaced with the string "Chair".

| Learner | Batch | VL-C (%) | L-C (%) | H-C (%) | VH-C (%) |
|---------|-------|----------|---------|---------|----------|
| GPT-3A-T0 | WS | 50.00 | 40.00 | 60.00 | 50.00 |
| | AS (I) | 50.00 | 60.00 | 60.00 | 60.00 |
| | AS (II) | 50.00 | 60.00 | 50.00 | 60.00 |
| GPT-3B-T0 | WS | 10.00 | 30.00 | 30.00 | 40.00 |
| | AS (I) | 70.00 | 50.00 | 20.00 | 60.00 |
| | AS (II) | 70.00 | 40.00 | 40.00 | 60.00 |
| GPT-3C-T0 | WS | 10.00 | 50.00 | 60.00 | 70.00 |
| | AS (I) | 80.00 | 50.00 | 70.00 | 60.00 |
| | AS (II) | 80.00 | 50.00 | 60.00 | 60.00 |
| GPT-3D-T0 | WS | 70.00 | 30.00 | 60.00 | 50.00 |
| | AS (I) | 70.00 | 40.00 | 50.00 | 60.00 |
| | AS (II) | 90.00 | 40.00 | 60.00 | 60.00 |
| GPT-3D-T0-S | WS | 40.00 | 0.00 | 40.00 | 20.00 |
| | AS (I) | 70.00 | 30.00 | 20.00 | 50.00 |
| | AS (II) | 90.00 | 40.00 | 60.00 | 30.00 |
| GPT-3D-T0-A | WS | 10.00 | 10.00 | 20.00 | 40.00 |
| | AS (I) | 70.00 | 50.00 | 20.00 | 60.00 |
| | AS (II) | 70.00 | 80.00 | 20.00 | 30.00 |
| GPT-3A-T1 | WS | 50.00 | 30.00 | 50.00 | 40.00 |
| | AS (I) | 40.00 | 40.00 | 50.00 | 60.00 |
| | AS (II) | 60.00 | 60.00 | 50.00 | 60.00 |
| GPT-3B-T1 | WS | 10.00 | 10.00 | 10.00 | 50.00 |
| | AS (I) | 80.00 | 40.00 | 50.00 | 60.00 |
| | AS (II) | 70.00 | 40.00 | 70.00 | 60.00 |
| GPT-3C-T1 | WS | 30.00 | 30.00 | 60.00 | 60.00 |
| | AS (I) | 70.00 | 50.00 | 60.00 | 60.00 |
| | AS (II) | 90.00 | 30.00 | 60.00 | 60.00 |
| GPT-3D-T1 | WS | 60.00 | 20.00 | 60.00 | 70.00 |
| | AS (I) | 50.00 | 20.00 | 50.00 | 60.00 |
| | AS (II) | 80.00 | 40.00 | 60.00 | 60.00 |

Table 8: Average accuracy obtained by the eight different GPT-3 configurations in the teaching task with concepts binned by complexity (from very low to very high) and batches, not including the test examples from the original experiment setting where the output is the empty string. Interpretation as in Table 7.

| Learner | Batch | VL-C (%) | L-C (%) | H-C (%) | VH-C (%) |
|---|---|---|---|---|---|
| GPT-3A-T0 | WS | 55.00 | 20.00 | 0.00 | 33.33 |
| | AS (I) | 57.5 | 40.00 | 0.00 | 33.33 |
| | AS (II) | 55 | 56.67 | 0.00 | 50.00 |
| GPT-3B-T0 | WS | 12.50 | 10.00 | 0.00 | 33.33 |
| | AS (I) | 75.00 | 46.66 | 33.33 | 50.00 |
| | AS (II) | 75.00 | 46.66 | 33.33 | 50.00 |
| GPT-3C-T0 | WS | 12.50 | 30.00 | 0.00 | 33.33 |
| | AS (I) | 90.00 | 30.00 | 16.67 | 50.00 |
| | AS (II) | 87.50 | 46.67 | 33.33 | 50.00 |
| GPT-3D-T0 | WS | 80.00 | 10.00 | 0.00 | 33.33 |
| | AS (I) | 80.00 | 30.00 | 33.33 | 50.00 |
| | AS (II) | 100.00 | 46.67 | 33.33 | 50.00 |
| GPT-3D-T0-S | WS | 42.50 | 0.00 | 33.33 | 33.33 |
| | AS (I) | 75.00 | 36.67 | 33.33 | 50.00 |
| | AS (II) | 100.00 | 53.33 | 50.00 | 50.00 |
| GPT-3D-T0-A | WS | 12.50 | 10.00 | 33.33 | 50.00 |
| | AS (I) | 75.00 | 70.00 | 33.33 | 50.00 |
| | AS (II) | 75.00 | 100.00 | 33.33 | 50.00 |
| GPT-3A-T1 | WS | 55.00 | 10.00 | 0.00 | 16.67 |
| | AS (I) | 45.00 | 20.00 | 0.00 | 50.00 |
| | AS (II) | 65.00 | 56.67 | 16.67 | 50.00 |
| GPT-3B-T1 | WS | 12.50 | 0.00 | 0.00 | 33.33 |
| | AS (I) | 87.50 | 30.00 | 33.33 | 50.00 |
| | AS (II) | 77.50 | 46.67 | 50.00 | 50.00 |
| GPT-3C-T1 | WS | 35.00 | 10.00 | 0.00 | 16.67 |
| | AS (I) | 80.00 | 30.00 | 0.00 | 50.00 |
| | AS (II) | 100.00 | 20.00 | 33.33 | 50.00 |
| GPT-3D-T1 | WS | 67.50 | 16.67 | 16.67 | 50.00 |
| | AS (I) | 55.00 | 10.00 | 33.33 | 50.00 |
| | AS (II) | 90.00 | 46.67 | 33.33 | 50.00 |

# 5 Explored Prompts

We present examples of the different kinds of prompts that have been used in the experiments with GPT-2 and GPT-3. These prompts were employed for two goals: test the performance on the learning task, and try to generate natural language explanations based on the examples. We tried different variations of prompts and several changes of alphabet. For instance, we did experiments where 0 and 1 were replaced by a and b, or "Apple" and "Banana". Some results reported in Tables 7 and 8.

Table 9: Examples of prompts used in the experiments with GPT-2 and GPT-3.

| System | Purpose | Prompt | Observations |
|---|---|---|---|
| GPT-2 | Test Performance | Input1: 0, Output1: 0; Input2: 00000, Output2: | Final prompt used for the experiment test. The numbering aims to logically connect and distinguish the different input-output pairs. |
| GPT-3 | Test Performance | Input: 0
Output: 0
Input: 00000
Output: | Final prompt used for the experiment test. In this case, instead of using ';' as the separator between pairs, we just included a line break. Also, numbering was omitted, following the prompt style guidelines for different problems presented in the OpenAI API. |
| GPT-3 | Test Performance | Input: 0
Output: 0
Input: 0,0,0,0,0
Output: | Prompt designed to consider the possible implicit byte-pair encoding of GPT-3. Discarded for the final experiments as results were worse than the "no separator" alternative (see Tables 7 and 8). |
| GPT-3 | Test Performance | Input: Apple
Output: Apple
Input: Apple Apple Apple Apple Apple
Output: | This prompt aimed to discover any performance difference if we changed the binary alphabet to represent the inputs and outputs. "0" is replaced with "Apple", "1" is replaced with "Banana" and "-" is replaced with "Chair". Discarded for the final experiments as results were not strictly different from those obtained with the original setting (see Tables 7 and 8). |
| GPT-3 | Test Performance | Input: a
Output: a
Input: aaaaa
Output: | This prompt also aimed to discover any performance difference if we changed the binary alphabet to represent the inputs and outputs. "0" is replaced with "a", "1" is replaced with "b" and "-" is not replaced. Discarded for the final experiments as results were not strictly different from those obtained with the original setting (see Tables 7 and 8). |
| GPT-3 | Test Performance | Q: If 0 changes to 0, what does 00000 change to?
A: | This prompt aimed to discover any performance difference if we changed the way in which we ask for answers, using a sentence rather than the explicit input-output labels. Discarded for the final experiments as results were not strictly different from those obtained with the original setting (see Tables 7 and 8). |

Table 10: Examples of the prompts used in the experiments with GPT-3 to generate natural language explanations.

| System | Purpose | Prompt | Observations |
|---|---|---|---|
| GPT-3 | Explanations | P: write a Python function that when given '0' as input, returns '0' Code: | As GPT-3 has demonstrated its capability to generate code from descriptions, the main idea of this prompt is to generate the explanations using Python scripts. |
| GPT-3 | Explanations | Truth: program('0')='0' Description: Print the first character of program's input Truth: program('0')='001' && program('01')='001' && program('1')='0' Description: If the first character of program's input is '0', print '001'; else, print '0' Truth: program('010')='1' && program('10')='0' Description: | In this prompt we give GPT-3 examples of how we would like it to behave, by representing the training examples input-output patterns and asking for a description related to the given truth. |
| GPT-3 | Explanations | My colleague asked me about the behaviour of these examples: An instance with the attribute 0 as '1', the attribute 1 as '1', the attribute 2 as '1', the attribute 3 as '1', the attribute 4 as '1' and the attribute 5 as '1' belongs to the class '1' (...) I rephrased it for him, in plain language: | This prompt aims to test the summarisation power of GPT-3 and use it for the explanation generation purpose. We write all the instances in natural language and ask GPT-3 to give an abstraction based on the examples. |
| GPT-3 | Explanations | Input1: 0 Output1: 0 Explanation: Print the first character of a string Input2: 00 Output2: 0 Input1: 01 Output1: 1 Explanation: | In a similar style to the prompt used for the experiment test, examples are provided to indicate the desired output, and finally we ask for the explanation using the teaching examples of interest. |

# 6 Detailed results and dispersion for systems with variability

Some results such as MH and Louise give one solution. Analysing variability is then more interesting in the variants of GPT where there is stochasticity (and several runs) or in cases such as humans, where we average the results for a population. Here we add the detailed results with their error bars (the standard deviation) for GPT-3A-T1, GPT-3B-T1, GPT-3C-T1, GPT-3D-T1, GPT-2-exp and humans (Figures 9, 10, 7, 8, 5, 6, 3, 4, 11, 12, 13 and 14).

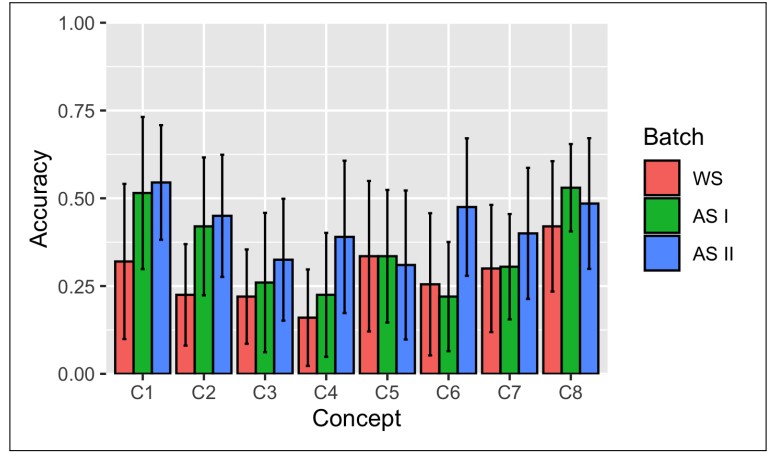

Figure 3: Performance mean and standard deviation bars for the 40 GPT-3A-T1 runs.

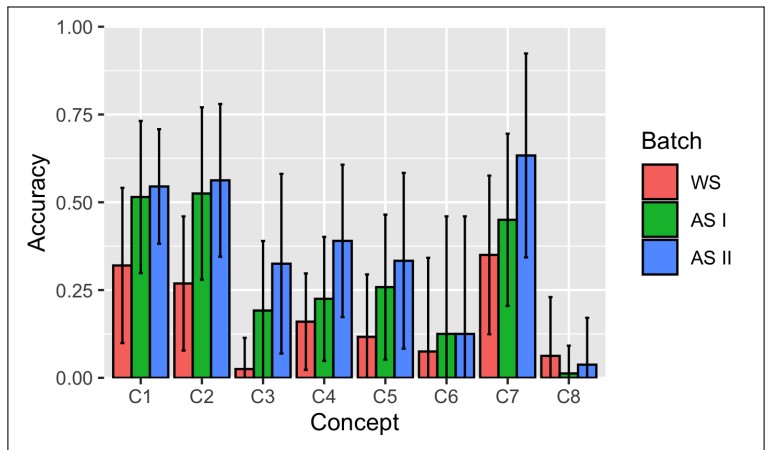

Figure 4: Performance mean and standard deviation bars for the 40 GPT-3A-T1 runs when the test examples from the original experiment setting with empty outputs are deleted.

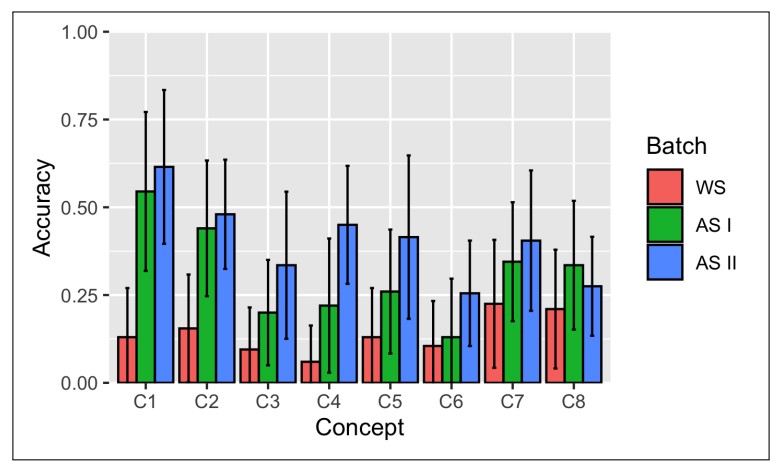

Figure 5: Performance mean and standard deviation bars for the 40 GPT-3B-T1 runs.

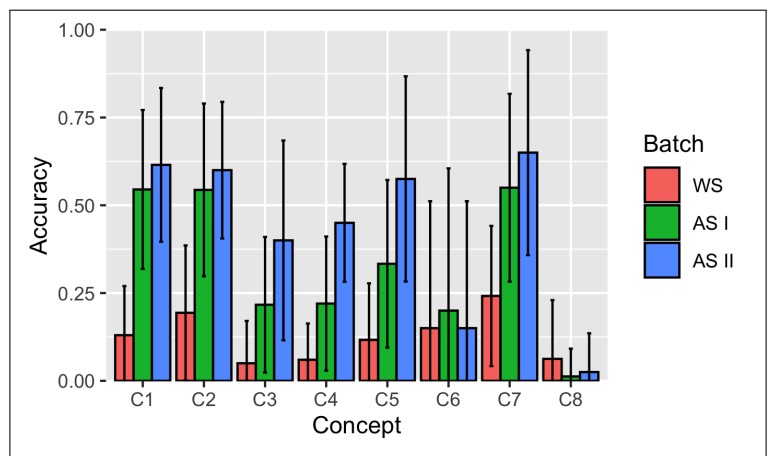

Figure 6: Performance mean and standard deviation bars for the 40 GPT-3B-T1 runs when the test examples from the original experiment setting with empty outputs are deleted.

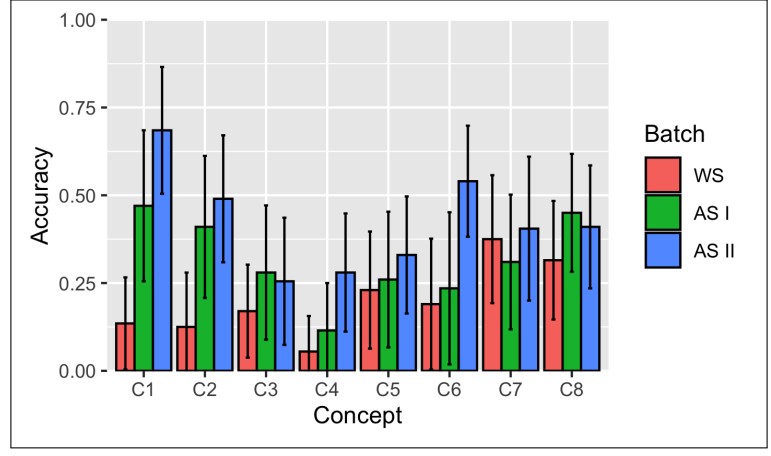

Figure 7: Performance mean and standard deviation bars for the 40 GPT-3C-T1 runs.

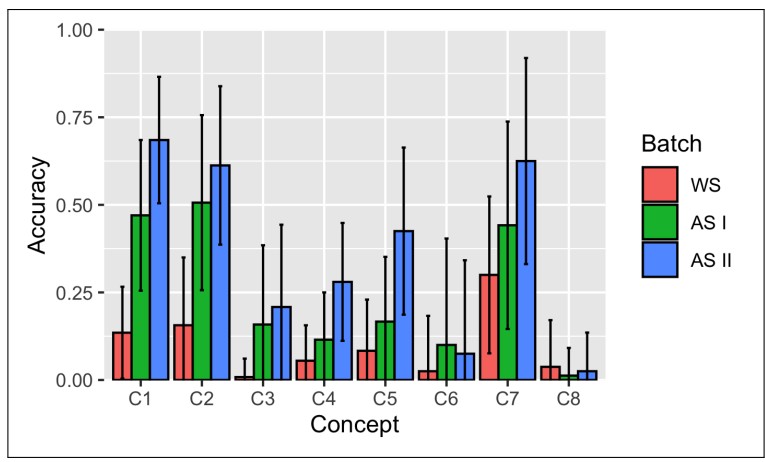

Figure 8: Performance mean and standard deviation bars for the 40 GPT-3C-T1 runs when the test examples from the original experiment setting with empty outputs are deleted.

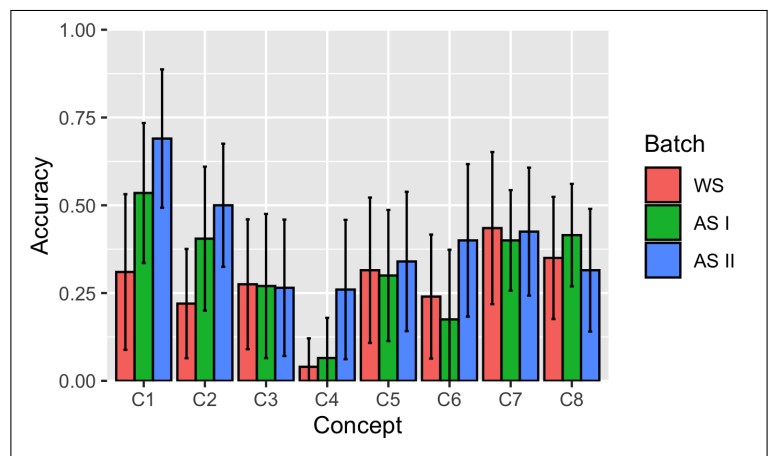

Figure 9: Mean and standard deviation bars for the 40 runs with GPT-3D-T1.

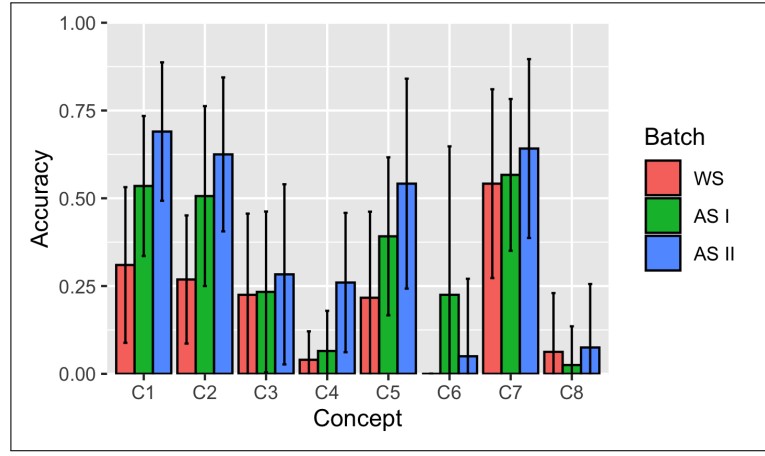

Figure 10: Performance mean and standard deviation bars for the 40 GPT-3D-T1 runs when the test examples from the original experiment setting where the output is the empty string are deleted.

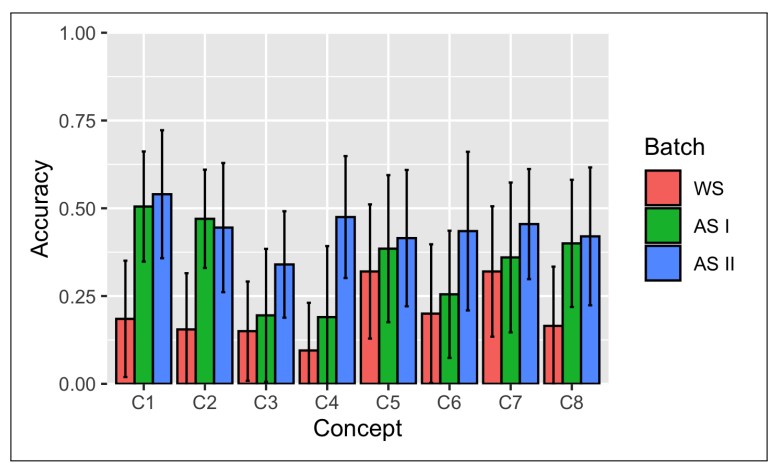

Figure 11: Performance mean and standard deviation bars for the 40 GPT-2 runs.

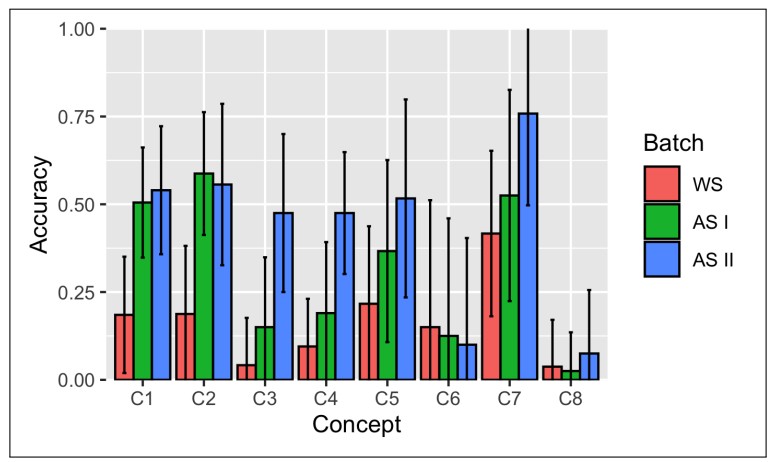

Figure 12: Performance mean and standard deviation bars for the 40 GPT-2 runs when the test examples from the original experiment setting with empty outputs are deleted.

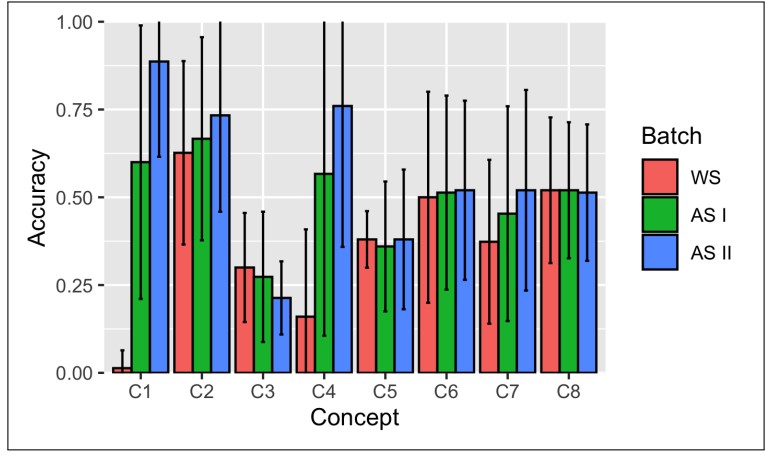

Figure 13: Performance mean and standard deviation bars for the 30 human participants performance.

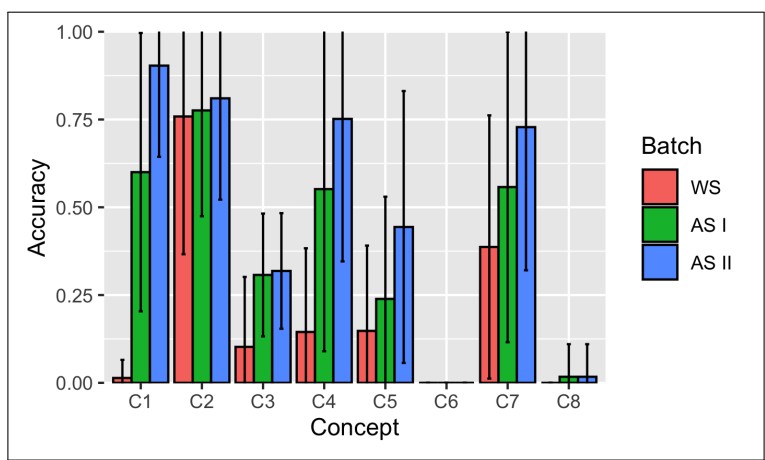

Figure 14: Performance mean and standard deviation bars for the 30 human participants when the test examples from the original experiment setting with empty outputs are deleted.

# 7 Additional Experiments: Concepts with Loops

To test GPT-3's capability of learning concepts with loops, we used four different concepts using loops of different complexities.

- Algorithmic Concept 1 (AC1), [o>] can be described as the concept "identity".
- Algorithmic Concept 1 (AC2), [oo>] can be described as the concept "duplicate each bit".
- Algorithmic Concept 1 (AC3), [>]+[<o] can be described as the concept "reverse".
- Algorithmic Concept 1 (AC4), [o>]<[<]>[o>] can be described as the concept "duplicate the string".

The results by the system GPT-3D-T0 are shown in Table 11. The learning performance is generally poor. Only the concept "identity" ([o>]) seems to be captured correctly.

Table 11: Average accuracy obtained by GPT-3D-T0 when learning algorithmic concepts with different complexities.(*) Witness sets for AC 2 and AC 4 are not obtained with the machine teaching setting explained in the paper, but hand-picked. (**) The learner was capable of learning the full concept with just two examples (WS + 1).

| Learner | Batch | AC 1 (%) | AC 2 (%) | AC 3 (%) | AC 4 (%) |
|---------|-------|----------|----------|----------|----------|
|         |       | [o>]     | [oo>]    | [>]+[<o] | [o>]<[<]>[o>] |
| GPT-3D-T0 | *WS | 0.00 | 0.00 | 0.00 | 0.00 |
|         | AS I  | **100.00 | 0.00 | 0.00 | 0.00 |
|         | AS II | 100.00 | 0.00 | 20.00 | 0.00 |

# 8 Additional Experiments: Concepts without Loops

As results showed a change on the accuracy decreasing trend when evaluating concepts with the highest complexity (VH-C, concepts with 7 or 8 instructions), Table 12 shows the results for some additional experiments with GPT-3D-T0, targeting concepts with 5 (H-C), 6 (V-HC) and 7 (V-HC) instructions:

- Additional Concept 5 (AC5), >ooooo, can be described as the concept "print the second character 5 times".
- Additional Concept 6 (AC6), o>ooo<o, can be described as the concept "print the first character once, then print the second character three times, and then print the first character once".
- Additional Concept 7 (AC7), o+oooo−o, can be described as the concept "if input is the empty string, print the empty string; else if input starts with '1', print '1'; else print '011110'".

What we see in these results is that the reference language is very important when using only the WS, and the results are very poor. For these more difficult concepts, AS I and especially AS II make a big difference. Table 13 incorporates all the concepts we have evaluated for GPT-3D-T0. With more concepts we see the trends are more clear, decreasing by complexity and increasing as more training data is provided. Figure 15 shows the mean accuracy for GPT-3D-T0 for the three teaching set batches and Figure 16 shows the mean accuracy for GPT-3D-T0 shown for the four complexity levels.

Table 12: Average accuracy obtained by GPT-3D-T0 when learning additional concepts AC5, AC6 and AC7 with H-C and VH-C complexities.

| Learner | Batch | AC5 (%) | AC6 (%) | AC7 (%) |
|---|---|---|---|---|
| | | >ooooo | o>ooo<o | o+oooo−o |
| GPT-3D-T0 | WS | 20.00 | 0.00 | 20.00 |
| | AS I | 80.00 | 60.00 | 20.00 |
| | AS II | 80.00 | 60.00 | 100.00 |

Table 13: Average accuracy obtained by GPT-3D-T0 using the averaging of the original 8 concepts, the four algorithmic concepts AC1, AC2, AC3, AC4 and the three additional concepts AC5, AC6, AC7. Note that the concept AC4 (with 9 instructions) has been included in the VH-C bin.

| GPT-3D-T0 | Batch | VL-C (%) | L-C (%) | H-C (%) | VH-C (%) |
|---|---|---|---|---|---|
| | | (C1-2) | (C3-4,AC1) | (C5-6,AC2,AC5) | (C7-8,AC3-4,AC6-7) |
| | WS | 80.00 | 6.67 | 5.00 | 14.44 |
| | AS (I) | 80.00 | 53.33 | 36.67 | 30.00 |
| | AS (II) | 100.00 | 64.44 | 36.67 | 46.67 |

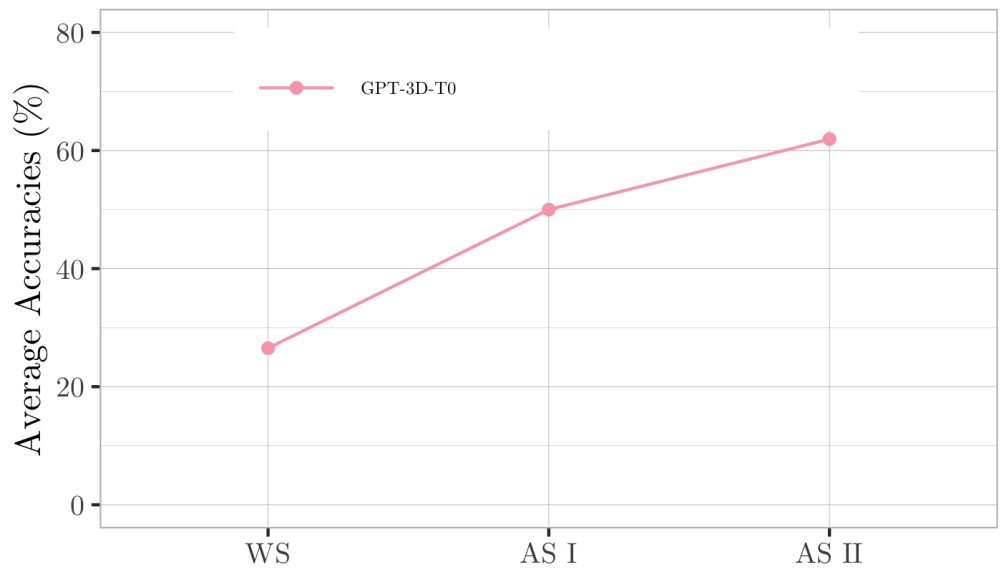

Figure 15: Mean accuracy for GPT-3D-T0 shown for the three teaching set batches with all the results in Table 13.

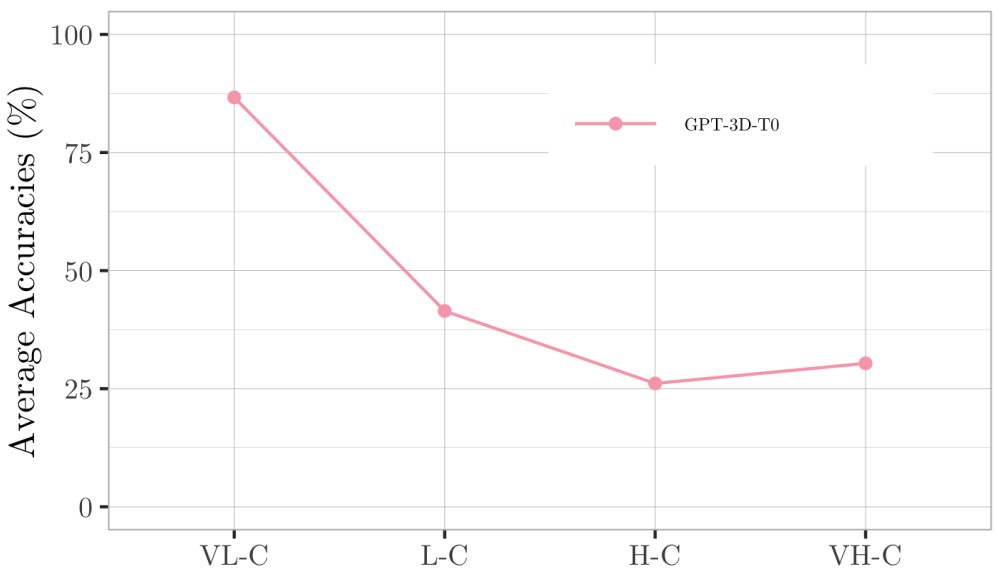

Figure 16: Mean accuracy for GPT-3D-T0 shown for the four complexity levels with all the results in Table 13.

## 9 Additional Experiments: Many More Concepts and Baselines

The choice of a small set of representative concepts was motivated because we wanted to include humans in our study. However, we can analyse the result with a much larger set of concepts to get more stable results. Some reviewers suggested to increase the number of examples in the test set as an alternative way of getting more robust results. However, as this would change the distribution of the test set, towards larger input-output pairs, we decided to look for more robust results by increasing the number of concepts only.

Another limitation for an arbitrary large number of concepts is that for each program size there is a finite number of programs, and for some sizes, this number is very small. For instance, there is

only one concept of program size 1, and only four concepts of program size 2. Table 14 shows the number of non-redundant concepts in P3 for each program length by our complexity groups VL-C, L-C, H-C, VH-C. As a result, we cannot increase the concepts for VL and we will find some limits in the number of concepts that we can explore for the L-C group.

Table 14: Number of non-redundant concepts in P3 according to their program length (the value for VH-C is an underestimate).

| Size | No. of programs |
|------|-----------------|
| VL-C | 5 |
| L-C | 47 |
| H-C | 366 |
| VH-C | 1475* |

We include the results of all GPT-3 configurations with zero temperature using the same methodology as in the paper (and test examples with empty outputs deleted) for a total of 5 VL-C + 47 L-C + 98 H-C + 150 VH-C concepts, getting the results shown in tables 15, 16, 17 and 18, which are very similar to our previous experiments.

Table 15: Average accuracy obtained by GPT-3A-T0 using 300 concepts (5 VL-C + 47 L-C + 98 H-C + 150 VH-C).

| GPT-3A-T0 | Batch | VL-C (%) | L-C (%) | H-C (%) | VH-C (%) |
|-----------|-------|----------|---------|---------|----------|
| | WS | 37.00 | 29.03 | 19.05 | 20.68 |
| | AS (I) | 48.00 | 40.53 | 26.02 | 24.30 |
| | AS (II) | 57.00 | 45.88 | 26.49 | 25.18 |

Table 16: Average accuracy obtained by GPT-3B-T0 using 300 concepts (5 VL-C + 47 L-C + 98 H-C + 150 VH-C).

| GPT-3B-T0 | Batch | VL-C (%) | L-C (%) | H-C (%) | VH-C (%) |
|-----------|-------|----------|---------|---------|----------|
| | WS | 20.00 | 25.20 | 15.75 | 13.75 |
| | AS (I) | 60.00 | 44.35 | 30.11 | 31.77 |
| | AS (II) | 65.00 | 49.92 | 34.94 | 28.36 |

Table 17: Average accuracy obtained by GPT-3C-T0 using 300 concepts (5 VL-C + 47 L-C + 98 H-C + 150 VH-C).

| GPT-3C-T0 | Batch | VL-C (%) | L-C (%) | H-C (%) | VH-C (%) |
|-----------|-------|----------|---------|---------|----------|
| | WS | 20.00 | 23.57 | 12.21 | 7.26 |
| | AS (I) | 51.00 | 37.68 | 19.89 | 18.81 |
| | AS (II) | 70.00 | 36.66 | 28.08 | 21.47 |

The results are much more stable than those in Table 6 (especially for the WS row), but nevertheless consistent. They also show a more consistent increase as more examples are given (last row over middle row and especially first row) and decrease for higher complexities.

Interestingly, the results for GPT-3B-T0 seem to be better than GPT-3D-T0. It is only for VL-C that the largest version of GPT-3 is better than the rest. This again suggests a non-monotonic relation between number of parameters and performance in our setting.

Finally, for the experiment with 300 concepts, we also included two baselines, *majority* and *random*. The *majority* baseline simply extracts the output string that is most common in the training

Table 18: Average accuracy obtained by GPT-3D-T0 using 300 concepts (5 VL-C + 47 L-C + 98 H-C + 150 VH-C).

| GPT-3D-T0 | Batch | VL-C (%) | L-C (%) | H-C (%) | VH-C (%) |
|-----------|-------|----------|---------|---------|----------|
| | WS | 37.00 | 20.05 | 6.71 | 10.02 |
| | AS (I) | 47.00 | 38.56 | 18.76 | 19.62 |
| | AS (II) | 75.00 | 46.70 | 29.55 | 22.35 |

examples (depending on the batch) and outputs the most frequent string as output. The *random* baseline simply generates an output string with probably $2^{-(l(s)+1)}$. For instance, the string '0' has probability 0.25, while the string '1011' has probability 0.015625. Table 19 shows the results for the majority baseline, and Table 20 shows the results for the random baseline (no need to separate by batches here as this baseline is independent on the seen examples). They are sufficiently low to exclude any significant effect in the analysis of results. This is why we only include them here in the supplementary material.

Table 19: Average accuracy obtained by the majority baseline for 300 concepts (5 VL-C + 47 L-C + 98 H-C + 150 VH-C).

| Majority | Batch | VL-C (%) | L-C (%) | H-C (%) | VH-C (%) |
|----------|-------|----------|---------|---------|----------|
| | WS | 6.67 | 3.56 | 3.54 | 3.65 |
| | AS (I) | 6.67 | 5.33 | 2.92 | 2.43 |
| | AS (II) | 6.67 | 5.33 | 2.92 | 2.43 |

Table 20: Average accuracy obtained by the random baseline for 300 concepts (5 VL-C + 47 L-C + 98 H-C + 150 VH-C).

| Random | VL-C (%) | L-C (%) | H-C (%) | VH-C (%) |
|--------|----------|---------|---------|----------|
| | 4.06 | 2.86 | 2.67 | 2.61 |

## 10   Additional Experiments: Not Using Machine Teaching

In the paper we have argued why machine teaching is the right approach to really know what the minimal set must be such that there is perfect identification between teacher and learner. Deviations from this optimal result would reflect a lack of alignment in the prior. However, there are many other ways of generating examples. Doing the generation less carefully would create training sets with redundancies. This would make it harder to understand factors such as the number and size of examples, and prior alignment.

We nevertheless may be interested in seeing the results when the examples are generated in a different way. However, in order to make an experiment of this kind comparable with the original machine teaching setting, we must keep some things constant. For instance, we could generate the training examples, the additional examples and the test set from a different distribution, but then we would compare both approaches against different test set distributions, especially because their sizes would be different. By using larger examples in the *test* set, we may get different results just because the outputs are larger and the corrections of guessing them by chance should be different (mapping 0000 into 1010 by chance is less likely than mapping 0 into 1 by chance). This is something that happens in structured prediction, unlike classification or regression. In order to consider a larger test set where the input strings may be larger, we force the output strings follow the *size distribution* of the original sets we used (e.g., examples such as mapping 0000 into 1).

More specifically, we want to compare the effect of the first batch, represented by the witness set (WS) chosen using machine teaching, with an alternative procedure of choosing the first batch as a

random set (let us call it RS 0). The most important thing for making this comparison fair is that both sets must have the same sizes. If we allow for larger sets for WS than RS 0, or vice versa, then this would be an unfair comparison, since one would have more bits than the other. With this constraint, there are still many choices to make the experiment informative. This is the configuration that we think is most informative. We kept the AS I, AS II and test sets as the original experiments, so that we only focus on the effect of the first batch, keeping everything else equal. We also kept the size $s$ of the original WS for each concept. Since this $s$ could give advantage to the old method (as they would never have overlap with AS I and AS II), we regenerated both approaches as follows.

- Experiment A: We generated the WS choosing from the witness sets of size $s$ in the teaching book, always checking that the examples are not in the test set.

- Experiment B: We generated a RS 0 choosing examples randomly of sets of size $s$, always checking that the set is neither in the witness sets for that concept in the teaching book nor its examples are in the test set. We also chose this configuration as otherwise for small complexities there would not be many instances to choose from, given a limit on the size.

In those cases where A or B could not complete the set, we used the original WS. Note that this configuration plays in favour of approach B, as we choose from a random distribution for Test and RS 0, while we choose from the teaching set distribution for WS. It is hard to think of a totally fair configuration, so we decided to choose against A. We generated 5 VL-C + 40 L-C + 40 H-C + 40 VH-C = 125 concepts, the same for both experiments. Table 21 shows the results for experiment A and Table 22 shows the results for experiment B.

Table 21: Average accuracy obtained by experiment A (using WS with machine teaching) for 5 VL-C + 40 L-C + 40 H-C + 40 VH-C = 125 concepts.

| GPT-3D-T0 (Experiment A) | Batch | VL-C (%) | L-C (%) | H-C (%) | VH-C (%) |
|---|---|---|---|---|---|
| | WS | 33.00 | 16.88 | 12.35 | 4.86 |
| | AS (I) | 55.00 | 34.17 | 14.66 | 18.67 |
| | AS (II) | 59.00 | 40.92 | 30.68 | 30.00 |

Table 22: Average accuracy obtained by experiment B (using RS 0 not using machine teaching) for 5 VL-C + 40 L-C + 40 H-C + 40 VH-C = 125 concepts.

| GPT-3D-T0 (Experiment B) | Batch | VL-C (%) | L-C (%) | H-C (%) | VH-C (%) |
|---|---|---|---|---|---|
| | **RS 0** | 18.00 | 15.08 | 9.53 | 15.52 |
| | AS (I) | 34.00 | 25.54 | 17.91 | 18.24 |
| | AS (II) | 75.00 | 40.45 | 34.70 | 31.33 |

The results of experiment A are very similar to those reported previously, as they both use a WS from the teaching book. When comparing the results between WS (from experiment A) and RS 0 (from experiment B), we see that WS gives better results, especially for low complexity. They seem to produce more informative examples with the same size.

However, the results for VH in experiment B are higher than expected. In this VH-C case, the value of $s$ is usually large and experiment B has more to choose from. This means there is more coincidence between the outputs of AS 0 and the test set. In other words, the test set does not try to cover a range of different cases as WS tries to do, but it is generated randomly with many repeated outputs that also benefit RS 0. Controlling for cases where the target output appears four or more times in the RS 0 gives a value of 0.067, which better fits the decreasing trend in experiment B. In any case, let us remember that we choose from a random distribution for the test set, which is used to evaluate RS 0 (sampled from the very same distribution) and WS (sampled from a different distribution, the witness set). Also note that it is not the goal of our paper to find the sample that gives the best results, but the one that best corresponds to the simplicity prior.

Overall, we think that these results are in agreement with the use of WS based on machine teaching being the most informative approach to represent the inductive bias used for P3, which translates partially to other representations, at least for concepts of small and medium complexity.

## 11  All the explanations produced by GPT-3

For generating the explanations in tables 23, 24, 25 and 26 we used the last prompt style presented in Table 10. This prompt presents a very similar style to that of the prompts used for the main experiments with GPT-3. The input and output labels are numbered, so GPT-3 knows when it should provide the output for the test example (the last one labelled as "Input1") and the explanation. Provided explanations for the teaching examples are basic explanations such as "return the first character" or if-else rules.

The explanations returned by GPT-3 are classified as "correct", "partially-correct" and "incorrect". An explanation is "partially-correct" when it explains part of the concept, but not the whole concept. An explanation is "incorrect" when it incorrectly explains a part or the whole concept.

Figure 3 of the main paper includes results about accuracy and size of the explanations for the learned concepts provided by humans. The concepts are those in Table 1 of this supplementary material.

Table 23: Explanations produced by GPT-3 for C1-C3

| Concept | Batch | Test | Explanation | Rating |
|---|---|---|---|---|
| C1 | WS | ('00000','0') | Explanation: if the input starts with 0, return 0; else, return 1 | Correct |
| | | ('11100','1') | Explanation: if the input starts with 1, return 0; else, return 1 | Incorrect |
| | | ('00111','0') | Explanation: return the input's second character | Incorrect |
| | | ('11010','1') | Explanation: return the input's second character | Incorrect |
| | | ('0010', '0') | Explanation: if the input starts with 0, return 0; else, return 1 | Correct |
| C1 | AS I | ('00000','0') | Explanation: return the input's second character | Incorrect |
| | | ('11100','1') | Explanation: return the input's second character | Incorrect |
| | | ('00111','0') | Explanation: if the input starts with 1, return 0; else, return 1 | Incorrect |
| | | ('11010','1') | Explanation: return the input's second character | Incorrect |
| | | ('0010', '0') | Explanation: return the input's second character | Incorrect |
| C1 | AS II | ('00000','0') | Explanation: return the input's second character | Incorrect |
| | | ('11100','1') | Explanation: return the input's second character | Incorrect |
| | | ('00111','0') | Explanation: return the input's second character | Incorrect |
| | | ('11010','1') | Explanation: return the input's second character | Incorrect |
| | | ('0010', '0') | Explanation: return the input's second character | Incorrect |
| C2 | WS | ('01011','1') | Explanation: return the input's second character | Correct |
| | | ('0101','1') | Explanation: return the input's second character | Correct |
| | | ('0010','0') | Explanation: if the input starts with 0, return 1; else, return 0 | Incorrect |
| | | ('100','0') | Explanation: if the input starts with 1, return 0; else, return 1 | Incorrect |
| | | ('1','') | Explanation: if the input starts with 1, return 0; else, return 1 | Incorrect |
| C2 | AS I | ('01011','1') | Explanation: return the input's second character | Correct |
| | | ('0101','1') | Explanation: return the input's second character | Correct |
| | | ('0010','0') | Explanation: return the input's second character | Correct |
| | | ('100','0') | Explanation: return the input's third character | Incorrect |
| | | ('1','') | Explanation: return the input's second character | Correct |
| C2 | AS II | ('01011','1') | Explanation: return the input's second character | Correct |
| | | ('0101','1') | Explanation: return the input's second character | Correct |
| | | ('0010','0') | Explanation: if the input starts with 0, return 1; else, return 0 | Incorrect |
| | | ('100','0') | Explanation: return the input's second character | Correct |
| | | ('1','') | Explanation: if the input is 1, return 0; else, return 1 | Incorrect |
| C3 | WS | ('00010','1') | Explanation: return the input's second character | Incorrect |
| | | ('110','') | Explanation: return the input's second character | Incorrect |
| | | ('00111','1') | Explanation: if the input is negative, return 0; else, return 1 | Incorrect |
| | | ('11000','') | Explanation: if the input is negative, return 0; else, return 1 | Incorrect |
| | | ('101','1') | Explanation: if the input starts with 1, return 0; else, return 1 | Incorrect |
| C3 | AS I | ('00010','1') | Explanation: if the input starts with 1, return 0; else, return 1 | Incorrect |
| | | ('110','') | Explanation: return the input's second character | Incorrect |
| | | ('00111','1') | Explanation: if the input is negative, return 0; else, return 1 | Incorrect |
| | | ('11000','') | Explanation: return the input's second character | Incorrect |
| | | ('101','1') | Explanation: return the input's second character | Incorrect |
| C3 | AS II | ('00010','1') | Explanation: if the input is 0, return 1; else, return 0 | Incorrect |
| | | ('110','') | Explanation: if the input starts with 1, return 0; else, return 1 | Incorrect |
| | | ('00111','1') | Explanation: return the input's second character | Incorrect |
| | | ('11000','') | Explanation: if the input is 0, return 1; else, return 0 | Incorrect |
| | | ('101','1') | Explanation: if the input is 1, return 0; else, return 1 | Incorrect |

Table 24: Explanations produced by GPT-3 for C4-C5

| Concept | Batch | Test | Explanation | Rating |
|---|---|---|---|---|
| C4 | WS | ('01011','011') | Explanation: return the input's second character | Incorrect |
| | | ('0101','011' | Explanation: return the input's second character | Incorrect |
| | | ('0010','011') | Explanation: return the input's second character | Incorrect |
| | | ('100','1') | Explanation: if the input starts with 0, return 1; else, return 0 | Incorrect |
| | | ('1','1') | Explanation: if the input starts with 0, return 1; else, return 0 | Incorrect |
| C4 | AS I | ('01011','011') | Explanation: return the input's third character | Incorrect |
| | | ('0101','011' | Explanation: return the input's second character | Incorrect |
| | | ('0010','011') | Explanation: return the input's second character | Incorrect |
| | | ('100','1') | Explanation: return the input's second character | Incorrect |
| | | ('1','1') | Explanation: return the input's second character | Incorrect |
| C4 | AS II | ('01011','011') | Explanation: return the input's second character | Incorrect |
| | | ('0101','011' | Explanation: return the input's second character | Incorrect |
| | | ('0010','011') | Explanation: return the input's second character | Incorrect |
| | | ('100','1') | Explanation: if the input is 0, return 1; else, return 0 | Incorrect |
| | | ('1','1') | Explanation: if the input is 0, return 1; else, return 0 | Incorrect |
| C5 | WS | ('01011','0') | Explanation: if the input starts with 1, return -; else, return 1 | Incorrect |
| | | ('110','1') | Explanation: if the input starts with 1, return 0; else, return 1 | Incorrect |
| | | ('0010','') | Explanation: if the input starts with 1, return 0; else, return 1 | Incorrect |
| | | ('101','1') | Explanation: if the input starts with 1, return 0; else, return 1 | Incorrect |
| | | ('1000','') | Explanation: if the input is negative, return 1; else, return 0 | Incorrect |
| | | ('01011','0') | Explanation: if the input is a string of 1s, return -; else, return 1 | Incorrect |
| C5 | AS I | ('110','1') | Explanation: if the input starts with 1, return 0; else, return 1 | Incorrect |
| | | ('0010','') | Explanation: if the input is 0, return 1; else, return 0 | Incorrect |
| | | ('101','1') | Explanation: if the input is a string of 1s, return -; else, return 1 | Incorrect |
| | | ('1000','') | Explanation: if the input is a 1, return 0; else, return 1 | Incorrect |
| | | ('01011','0') | Explanation: return the input's second character | Incorrect |
| C5 | AS II | ('110','1') | Explanation: return the input's second character | Incorrect |
| | | ('0010','') | Explanation: return the input's second character | Incorrect |
| | | ('101','1') | Explanation: if the input starts with 1, return 0; else, return 1 | Incorrect |
| | | ('1000','') | Explanation: return the input's second character | Incorrect |

Table 25: Explanations produced by GPT-3 for C6

| Concept | Batch | Test | Explanation | Rating |
|---|---|---|---|---|
| C6 | WS | ('01','00') | Explanation: if the input is 0, return 10; else, return -1 | Incorrect |
| | | ('0000','') | Explanation: if the input is 0, return 10; else, return -1 | Incorrect |
| | | ('00011','') | Explanation: if the input is 0, return 10; else, return - | Incorrect |
| | | ('0011','') | Explanation: if the input is 0, return 10; else, return - | Incorrect |
| | | ('1000','') | Explanation: if the input is 0, return 10; else, return -1 | Incorrect |
| C6 | AS I | ('01','00') | Explanation: if the input is 0, return 10; else, return -1; if the input is 1, return -1; else, return 1; if the input is 10, return -1; else, return 1; if the input is 11, return -1; else, return 1; | Incorrect |
| | | ('0000','') | Explanation: if the input is 0, return 0; else, return 1 | Incorrect |
| | | ('00011','') | Explanation: if the input is 0, return 10; else, return -1 | Incorrect |
| | | ('0011','') | Explanation: if the input is 0, return 1; else, return 0 | Incorrect |
| | | ('1000','') | Explanation: if the input is 0, return 10; else, return -1 | Incorrect |
| C6 | AS II | ('01','00') | Explanation: if the input is 0, return 10; else, return -1 | Incorrect |
| | | ('0000','') | Explanation: if the input is 0, return 10; else, return -1; if the input is 1, return 0; else, return -1; if the input is 2, return -1; else, return 1; if the input is 3, return -1; else, return 1; if the input is 4, return 1; else, return -1; if the input is 5, return -1; else, return 1; if the input is 6, return 1; else, return -1; if the input is 7, return -1; else, return 1; if the input is 8, return 1; else, return -1; if the input is 9, return -1; else, return 1; | Incorrect |
| | | ('00011','') | Explanation: if the input is 0, return 10; else, return -1; if the input is 1, return 0; else, return -1; if the input is 2, return -1; else, return 1; if the input is 3, return -1; else, return 1; if the input is 4, return 1; else, return -1; if the input is 5, return -1; else, return 1; if the input is 6, return 1; else, return -1; if the input is 7, return -1; else, return 1; if the input is 8, return 1; else, return -1; if the input is 9, return -1; else, return 1; if the input is 10, return -1; else, return 1; if the input is 11, return 1; else, return -1; if the input is 12, return -1; else, return 1 | Incorrect |
| | | ('0011','') | Explanation: if the input is 0, return 10; else, return -1; if the input is 1, return 0; else, return -1; if the input is 2, return -1; else, return 1; if the input is 3, return -1; else, return -1; if the input is 4, return -1; else, return 1; if the input is 5, return -1; else, return -1; if the input is 6, return -1; else, return 1; if the input is 7, return -1; else, return -1; if the input is 8, return -1; else, return 1; if the input is 9, return -1; else, return -1; if the input is 10, return -1; else, return 1; if the input is 11, return -1; else, return -1; if the input is 12, return -1 | Incorrect |
| | | ('1000','') | Explanation: if the input is 0, return 10; else, return -1; if the input is 1, return 0; else, return -1; if the input is -1, return 1; else, return -1; if the input is -1, return 1; else, return -1; if the input is -1, return 1; else, return -1; if the input is -1, return 1; else, return -1; if the input is -1, return 1; else, return -1; if the input is -1, return 1; else, return -1; if the input is -1, return 1; else, return -1; if the input is -1, return 1; else, return -1; if the input is -1, return 1; else, return -1; if the input is -1, return 1; else, return -1; if the input is | Incorrect |

Table 26: Explanations produced by GPT-3 for C7-C8

| Concept | Batch | Test | Explanation | Rating |
|---|---|---|---|---|
| C7 | WS | ('01011','1') | Explanation: if the input is 0, return -1; else, return 0 | Incorrect |
| | | ('0000','0') | Explanation: if the input is 0, return 0; else, return -1 | Incorrect |
| | | ('00000','0') | Explanation: if the input is 0, return 0; else, return -1 | Incorrect |
| | | ('100','') | Explanation: if the input starts with 0, return -; else, return 0 | Incorrect |
| | | ('1000','') | Explanation: if the input is 0, return -1; else, return 0 | Incorrect |
| C7 | AS I | ('01011','1') | Explanation: return the input's third character | Incorrect |
| | | ('0000','0') | Explanation: return the input's second character | Incorrect |
| | | ('00000','0') | Explanation: return the input's third character | Incorrect |
| | | ('100','') | Explanation: return the input's third character | Incorrect |
| | | ('1000','') | Explanation: return the input's second character | Incorrect |
| C7 | AS II | ('01011','1') | Explanation: return the input's third character | Incorrect |
| | | ('0000','0') | Explanation: return the input's second character | Incorrect |
| | | ('00000','0') | Explanation: return the input's third character | Incorrect |
| | | ('100','') | Explanation: return the input's third character | Incorrect |
| | | ('1000','') | Explanation: return the input's second character | Incorrect |
| C8 | WS | ('10101','') | Explanation: if the input is negative, return the input; else, return the negation of the input | Incorrect |
| | | ('11101','') | Explanation: if the input is negative, return 0; else, return 1 | Incorrect |
| | | ('00000','1') | Explanation: if the input is 0, return 1; else, return 0 | Incorrect |
| | | ('0011','1') | Explanation: if the input is negative, return the input; else, return -1 | Incorrect |
| | | ('1111','') | Explanation: if the input is negative, return 1; else, return the input | Incorrect |
| C8 | AS I | ('10101','') | Explanation: if the input is -, return 1; else, return the input's second character | Incorrect |
| | | ('11101','') | Explanation: if the input is -, return 1; else, return the input | Incorrect |
| | | ('00000','1') | Explanation: if the input is 0, return 1; else, return 0 | Incorrect |
| | | ('0011','1') | Explanation: if the input is -, return 1; else, return 0 | Incorrect |
| | | ('1111','') | Explanation: if the input is -, return 1; else, return the input | Incorrect |
| C8 | AS II | ('10101','') | Explanation: if the input is 0, return 1; else, return 0 | Incorrect |
| | | ('11101','') | Explanation: if the input is 0, return 1; else, return 0 | Incorrect |
| | | ('00000','1') | Explanation: if the input is 0, return 0; else, return 1 | Incorrect |
| | | ('0011','1') | Explanation: if the input is 0, return 1; else, return 0 | Incorrect |
| | | ('1111','') | Explanation: if the input is 0, return 1; else, return 0 | Incorrect |

## 12 Software licences

Table 27 shows the software used for the experimental results[2]. In those cases where the use of the software required citing the paper (e.g., Louise), we have done so. Following the recommendations of the OpenAI API[3], we communicated by email the publication of this paper to OpenAI.

Table 27: Licence of the software used in this work

| System | Licence |
|---|---|
| MagicHaskeller | BSD-3-Clause.[4] |
| Louise | No explicit licence (Default copyright laws apply). |
| GPT-2 | Modified MIT License.[5] |
| GPT-3 | Proprietary. Accessed through API. |

## 13 Hardware specifications

Table 28 shows the infrastructure that we have used. The teaching book was reused from a previous work Telle et al. (2019). The amount of compute for the first three systems (MagicHaskeller, Louise and GPT-2) was quite small compared to the resources that were used for GPT-3.

Table 28: Specifications of the hardware used in this work

| Experiments | Specifications |
|---|---|
| MagicHaskeller | Intel(R) Xeon(R) CPU E5-2609 v4 @ 1.70GHz, with 252GB RAM DIMM Synchronous 2400 MHz. |
| Louise | Intel Core i5 2GHz Quad-Core, with 16GB 3733MHz LPDDR4X RAM. |
| GPT-2 | Intel(R) Xeon(R) CPU @ 2.20GHz, Tesla T4, with 12.69GB RAM (Google Colab). |
| GPT-3 | Unknown (approximately 3,000,000 API tokens). |

## 14 Formal Definition of Teaching Size and the Teaching Book

Following Telle et al. (2019), we consider a possibly infinite example space $X$ and a possibly infinite concept class $C$ consisting of concepts over $X$. Examples are defined as pairs $(i, o)$ and concepts as functions mapping inputs to outputs. An example set $S = \{(i_1, o_1), \ldots, (i_k, o_k)\}$ is just a finite set of input-output pairs, used as witness for the teaching process. Given a concept $c \in C$ and an example set $S$, we say that $c$ satisfies $S$, denoted by $c \vDash S$, if $c(i) = o$ for all the pairs $(i, o)$ in $S$. The empty set is satisfied by all the concepts. We define an encoding function $\delta$ as the number of bits needed to encode $S$, then $\delta(S)$ represents the size of $S$. The teaching size is defined as:

$$TS(c) = \min_{S}\{\delta(S) : \{c\} = \{c' \in C : c' \vDash S\}\}$$

A concept $c$ can be implemented in a language $L$ by zero, one or more programs. A program $p$ captures concept $c$ iff for every $S$ we have that $p \vDash S$ iff $c \vDash S$. The equivalence class for $c$ is $Class_L(c) = \{p : \forall S, p \vDash S \iff c \vDash S\}$. The learner $\Phi$ can be seen as a function mapping sets to programs. After this, the teaching size for $\Phi$ can be reformulated as follows:

$$TS(c) = \min_{S}\{\delta(S) : \Phi(S) \in Class_L(c)\}$$

---

[2]The complete code for the experiments can be found at https://github.com/gonzalojaimovitch/think-big-teach-small

[3]https://beta.openai.com/policies/sharing-and-publication

[4]MagicHaskeller Licence: https://hackage.haskell.org/package/MagicHaskeller-0.9.6.8/src/LICENSE

[5]GPT-2 Licence: https://github.com/openai/gpt-2/blob/master/LICENSE

Priors can be used as more than one concept might be consistent with one given set $S$. One natural choice is simplicity, which can be formally defined as the size of the concepts. Let $l(p)$ be the length in bits of a program $p$ in $L$ using an appropriate encoding. Let $\prec$ be the total order of programs ordered by $l$, where shorter programs precede longer ones. In case of equal $l$, ties are broken lexicographically (more on this in section 16).

So the teaching size and the learner can now be expressed as:

$$TS_l(c) = \min_S \{\delta(S) : \Phi_l(S) \in Class_L(c)\} \quad \text{with} \quad \Phi_l(S) = arg\min_p^{\prec}\{l(p) : p \vDash S\}$$

From these equations the concept of the *teaching book* derives easily. The *teaching book* is a list consisting of entries in the form $(w, p)$ with $w$ being the optimal witness set and $p$ the smallest program compatible with $w$. The book is ordered by the encoding size $\delta$ for $w$. The algorithm enumerates on the size of $w$ and looks for the shortest program $p$ in each case, and does not include $\langle w, p \rangle$ if $p$ has appeared before. All this ensures that no $w$ is repeated and no $p$ is repeated. This setting allows to prove theoretically and empirically that it is possible to find witness sets for explaining concepts whose size is smaller than the programs they identify, "an illuminating justification of why machine teaching from examples makes sense at all" (Telle et al., 2019).

## 15 Distilling Simplicity, and Occam's Razor for Few-shot Inference

In this paper we analyse whether Occam's razor, i.e., a simplicity prior, is *distilled* by language models when used for few-shot inference. In this section we review the key elements of Occam's razor in machine learning, what priors are and how knowledge that is distilled can change these priors when performing inference. We want to properly clarify why we mean by having a simplicity prior in few-show inference, and the possibility and contingency of the main hypothesis we evaluate empirically in this paper.

Occam's razor is a quintessential concept in inductive inference and philosophy of science, and it is ubiquitous in statistics and machine learning, explicitly —through criteria such as the MML/MDL principles (Wallace and Boulton, 1968; Rissanen, 1983) and the use of regularisation terms— or implicitly —in the optimisation algorithms or parameter-function maps in many methods, including deep learning (Valle-Perez et al., 2019; Shah et al., 2020). The no free lunch theorems (Wolpert, 1996) would make learning impossible when considering infinite function classes if we assumed block uniformity in the data. Consequently, the assumptions of these theorems are thought to be false in practice. Indeed, there are strong theoretical and empirical reasons to believe that all hypotheses should not be equally likely a priori, and a strong Occam's razor should be assumed in any kind of inductive inference. See, e.g., (Solomonoff, 1964; Dingle et al., 2018) for some relevant pointers.

Occam's razor must be well understood, though, especially in the context of overparametrised models such as deep learning. A naive interpretation of Occam's razor would suggest that massive neural network architectures are against Occam's razor, but this is not necessarily true. For instance, in the particular case of transformers (Vaswani et al., 2017) and large language models (Brown et al., 2020), it is not that shortest 'models' are preferable, because the number of parameters as 'size' is not necessarily comparable, as their weights can have more or less information after training. For instance, a naive interpretation would be that, given some training data $X$, for which two models $M_1$ and $M_2$ have the same loss ($\psi(M_1(X)) = \psi(M_2(X))$, with $\psi$ being a metric of performance), model $M_1$ would be preferable over $M_2$ if size($M_1$) < size($M_2$). Rather, the correct interpretation is that $M_1$ is preferred over $M_2$ when both have the same loss ($\psi(M_1(X)) = \psi(M_2(X))$) if $K(M_1) < K(M_2)$, with $K$ being the Kolmogorov complexity of the *function represented by the model*. In other words, the prior applies to the complexity of the functions represented by $M_1$ and $M_2$.

The choice of Occam's razor as a prior for *learning*, as discussed above, is standard and well-known. The question comes when these language models and other pre-trained models are used for few-shot inference, usually referred to as 'few-shot learning'. It is important to clarify that the weights of the language model do not change at this deployment stage, so there is no further tuning or adaptation. Consequently, in strict technical terms, unlike other interpretations of 'few-shot learning' (Perez et al., 2021) we should be careful when using the term 'learning'. When applying these models we are actually doing inference, but not an *inductive* inference that changes weights or creates a new representation or model. This also holds for language models, as pre-trained models. However,

and this is precisely what we analyse in this paper, after being trained with such large and diverse corpora they acquire a vast amount of knowledge that is really what *conditions* the continuations (the probability of the next token). Inference (and hence text generation) is performed by choosing the tokens that best accommodate the model. So, what is really happening when a language model is given an intrinsic pattern such as **ababab**...? We cannot really say the model is retrieving the pattern, since in many cases good performance is achieved even when it is unlikely that the pattern appeared previously in the training data. Consequently, there must be some *abstract representations* that capture a preference for simple patterns and are triggered by the prompt. There is already strong evidence that transformers can distil structural bias from raw data (Warstadt and Bowman, 2020).

It is not a coincidence that in this paper we compare language models with Louise, which is actually an inductive logic programming system similar to Metagol (Muggleton et al., 2014), a paradigm that uses higher-order meta-rules that are instantiated by abduction to the given evidence. This emphasises that, in the context of transformers, few-shot 'learning' is actually abduction (seen as some kind of conditioning over previous knowledge) for some given abstract representations. When we ask the question of whether Occam's razor is distilled by language models, we are asking the question of whether a very abstract representation, such as a preference for simplicity, can be distilled during learning (Wu et al., 2021). This abstraction must be understood in a scale that is different from the first-order vs higher-order distinction of declarative languages. We use the term 'distillation', in a somewhat related way to 'knowledge distillation' (Gou et al., 2021), used for simplified models that capture the essence of the acquired model. Here, when the model is applied for inference, we consider Occam's razor as a kind of very abstract knowledge that acts as a prior for abduction.

Clarified all this, let us now try to formalise what it means to have an Occam's razor prior for few-shot inference. Consider we have a pre-trained model $M$ to make continuations of input strings (or prompts) $s$. For the sake of the argument, let us consider $M$ to be deterministic (e.g., by choosing the continuation with highest probability), then the continuation to $s$ is given by $M(s)$. Now consider that we can wrap a set of pairs of inputs and outputs $S$ and a final input $i$ into a string $s$ and feed $M$ with it so that the continuation is the output for $i$. In this few-shot setting, which is exactly the one used in the paper, $M$ *has distilled a simplicity prior* in language $L$ in a few-shot inference context as much as it minimises $\mathbb{E}_{s,i}[\delta(M(\langle s, i \rangle), p(i))]$ where $\delta$ is a divergence metric, $s$ wraps $S$, $i$ is the input of test instance and $p$ is the shortest program for $S$. Using the notation of section 14 here in the supplementary material, we have that $p = \Phi_l(S)$, where $\Phi_l$ is simply the learner in our machine teaching setting, guided by Occam's razor, i.e., $\Phi_l(S) = \arg\min_p^\prec \{l(p) : p \vDash S\}$.

While we talk about distilling simplicity or Occam's razor in general, this is always relative to a language $L$. The invariance theorem (see, e.g., Li and Vitányi, 2008) ensures that the use of different (optimal) languages gives at most a constant difference that only depends on these two languages. While these constants may be very large, some level of agreement is usually observed even for very different representations.

It is at this point that we can properly understand if the hypothesis that gives title to this paper and that we evaluate experimentally is tenable and contingent, otherwise it would be pointless to do experiments other than corroboration. If Occam's razor happens in many of the patterns in the data used to train a language model $M$, the metric $\psi$ will favour abstractions that are aligned with that razor. As known with some other deep learning architectures, these more complex abstractions can only be made when sufficient network layers and data are available, so it seems reasonable to expect that larger models may have been able to distil Occam's razor in higher degrees than smaller models.

Once established that the hypothesis is possible, is it contingent? Do neural networks favouring Occam's razor when training imply that they have to distil Occam's razor when used during deployment? If this was necessary (not contingent) there would not be much insight in performing the experiments. The necessity of the hypothesis is easy to reject by a counterexample. For instance, imagine the class $\mathcal{C}$ of all computable functions that map binary strings into binary strings such that the output string always has parity 1 (the number of 1s in the string is an odd number). Imagine that a model $M$ has been trained with a strong Occam's razor prior from the whole class $\mathcal{C}$. Now imagine that we choose each of these functions in isolation, and 'learn' separate models $M_k$ using Occam's razor for each $c_k \in \mathcal{C}$. This $M_k$ would frequently make wrong predictions, as it cannot know that parity-0 outputs are not allowed. For instance, given the sample $S = \{\langle 1, 1 \rangle, \langle 100, 100 \rangle, \langle 0111, 0111 \rangle, \}$, Occam's razor applied to this $S$ in isolation would likely

learn to identity function $id$, but $id \notin \mathcal{C}$. Consequently, there are situations whether Occam's razor for learning does not necessarily entail Occam's razor for few-shot inference.

Investigating whether this is the case for language models is hence both tenable and contingent, which means that any outcome is informative. This is why it is worth being explored experimentally, as we do in this paper.

## 16 Program Length Ties and Lexicographic Preference

In the machine teaching setting we are using, the witness set is calculated as the smallest set of examples (in overall string size) that makes the learner output a program compatible with the concept. The ideal learner the teacher has in mind is an enumerator, following the simplicity prior for programs. However, in case of a tie, a lexicographic order is used for programs.

This lexicographic order (on the programs) has a few consequences. First, it includes an arbitrary choice that is not strictly based on simplicity and makes identification harder. Second, we also use the lexicographic order when two witness sets have the same size (and both would be sufficient for the concept). In particular, this lexicographic order (on the strings) has an additional effect on language models, as we are using them in a few-shot learning scenario. In the complexities VL and L we usually find witness sets such $\{$ "" $\rightarrow$ "0111"$\}$, because the empty string comes lexicographically before other examples of the same size as $\{$ "01" $\rightarrow$ "10"$\}$. Also, it would come before $\{$ "0" $\rightarrow$ "1", "1" $\rightarrow$ "1"$\}$ too. In many cases, the number of examples matters more than the size of examples for language models in a few-shot inference scenarios, because this consolidates the pattern for the prompt, so that the continuations are more reliable.

Overall, this suggests that future studies should put more attention to the lexicographic order in the machine teaching scenario (for programs and examples)