# OpenReview forum: "Think Big, Teach Small: Do Language Models Distil Occam’s Razor?"
_NeurIPS.cc/2021/Conference — NeurIPS 2021 Poster_

### Official Review · Reviewer_HZ79 · 2021-07-03

**Rating:** 6
**Confidence:** 3

**Summary:**

This paper empirically compares humans, language models, and inductive programming systems through the framework of machine teaching. Concepts and example sets are selected using a simplicity prior with reference to the P3 language. The main experiments present the example sets to the humans or machines and ask them to infer the corresponding concept. This setup leads to several experimental questions regarding the alignment between humans and the computational approaches and the extent to which language models like GPT have distilled a simplicity prior.

**Limitations And Societal Impact:**

See “concerns” above.


**Main Review:**

Note: I am not an NLP researcher so I cannot comment on this work’s significance to that community.


Positive comments:
* The basic premise of this paper, to study the intersection of machine teaching and language models, is very interesting, and as far as I know, novel, and I expect there will be follow up work that continues this line of inquiry. I would therefore give this paper high marks in terms of originality and significance.
* The paper is extremely well written at both a sentence level and a high level. I found the whole paper very enjoyable to read and would give high marks in terms of clarity.
* The experiments are clear and thorough and the supplemental results are extensive.
* The overall contribution of the paper is large, clearly requiring an extensive effort, making it well suited for NeurIPS in terms of scope.
* I commend the authors for reporting the experimental question Q6 even though they were not able to answer it affirmatively. (As an aside, the teacher-student conversation in supplementary materials 8, while a “failure”, is remarkably coherent.)


Concerns and questions:
* My biggest concern is that nearly all of the results in the paper are based on the same 8 concepts. It is very possible that some of the empirical findings are specific to quirks in these 8 concepts. It makes sense that the human experiments needed a limited set, but it would greatly strengthen the paper if additional machine-only results were added with results averaged over hundreds or thousands of concepts.
* I am also concerned about the relative mismatch between the average accuracies of humans (e.g. Figure 2b) versus their ability to provide correct explanations (Figure 3a). Should we conclude that humans are sometimes guessing correctly? Or that there are some quirks in the concepts, where the examples are suggestive of some other concept that is wrong, but happens to align with the test examples? Also, the relatively low rate of explanation correctness across the board suggests that this task may just be too difficult for humans (it is certainly difficult for me, looking at the examples).
* For the “accuracy” results, if I were to guess 2 out of 5 tests correctly for a single concept, would that contribute 0.4 or 0.0 to the accuracy? Please clarify in the text.
* There are a lot of interesting results and points raised in this paper, which is mostly a strength, but there are parts where the narrative feels unfocused and the main takeaways feel unclear.
* I am wondering if the inclusion of the inductive programming methods (Louise and MH) are more distracting than informative, at least in the way that they are presented now, as first-class citizens alongside the GPT and human results. My understanding is that the behavior of these methods is much less mysterious than that of GPT or humans -- if the background knowledge given to the inductive programming methods were perfectly aligned with P3, then we would see perfect agreement in the results. So really the results for these methods are just a function of the background knowledge. I’m not sure what we are meant to take away from these results in particular.
* The link in footnote 1 (https://bit.ly/Code3357) cannot be accessed (permission error)
* For both humans and GPT, I would be curious to see if the results are robust to changes in the syntax. For example, instead of 0 and 1, what about a and b, or apple and banana, # and %, etc.
* When presenting the non-human approaches with WS, then WS + AS I, then WS + AS I + AS II, are the methods “reset” after each phase? For example, with GPT-3, does it see the full history of phases, or is just given each phase independently? I am guessing the latter. If so, is it possible that the humans have an inherent advantage over the machines? Intuitively, from the sequence, the humans could infer the relative complexity of the examples in each set, giving some additional hints about the target concept.

**Time Spent Reviewing:**

6

---

> ### Author Response · Authors · 2021-08-09
> **Response to reviewer "HZ79"**
>
> --- R: “My biggest concern is that nearly all of the results in the paper are based on the same 8 concepts. It is very possible that some of the empirical findings are specific to quirks in these 8 concepts. It makes sense that the human experiments needed a limited set, but it would greatly strengthen the paper if additional machine-only results were added with results averaged over hundreds or thousands of concepts.” ---
> Apart from the original 8 concepts, which we also cover in the human study, the supplementary material already included four extra algorithmic concepts and three additional complex concepts, partly motivated by a request from the IJCAI reviewers. The overall results for these 15 concepts support the original findings, with the trends becoming more clear and less dependent on the original choice of 8 concepts. But as more than one reviewer is suggesting to further increase this number, we will consider additional sets for the language models in the following versions of the paper. Hundreds or thousands of concepts would be ideal, but there are a couple of issues here. First, the number of low-complexity concepts is limited, that’s why we tend to extend the numbers with high-complexity concepts instead. Second, there’s a by-token cost according to our license of OpenAI’s API, so we will do our best to balance the economic resources we have (and the overall energy consumption of the experiments) with the information we really need to add even more extra support our findings. That’s in line with prioritising those cases where we had more variability (see sup. section 5). We think that we can find a pertinent increase in the number and choice of new concepts before the final version, also keeping the code available on github so that other researchers can replicate the results with these and further concepts, and other language models.
>
> --- R: “I am also concerned about the relative mismatch between the average accuracies of humans (e.g. Figure 2b) versus their ability to provide correct explanations (Figure 3a). Should we conclude that humans are sometimes guessing correctly? Or that there are some quirks in the concepts, where the examples are suggestive of some other concept that is wrong, but happens to align with the test examples? Also, the relatively low rate of explanation correctness across the board suggests that this task may just be too difficult for humans (it is certainly difficult for me, looking at the examples).” ---
> As the reviewer pointed out, many of the concepts are difficult to learn from very few examples. In some cases, humans are able to understand partially the behaviour of the concept and in this way they are able to guess correctly some of the test examples. However, when comparing the explanations with the target concept, we were much stricter in giving the correct correspondence. Additionally, many humans did not provide (meaningful) explanations.
>
> --- R: “For the “accuracy” results, if I were to guess 2 out of 5 tests correctly for a single concept, would that contribute 0.4 or 0.0 to the accuracy? Please clarify in the text.” ---
> We account every example independently, therefore the contribution would be 0.4. We will clarify this.
>
> --- R: “There are a lot of interesting results and points raised in this paper, which is mostly a strength, but there are parts where the narrative feels unfocused and the main takeaways feel unclear.” ---
> We will improve the narrative in the next versions, focusing on the original questions and the contributions. We will follow the valuable comments raised by the reviewers.”
>
> --- R: “I am wondering if the inclusion of the inductive programming methods (Louise and MH) are more distracting than informative, at least in the way that they are presented now, as first-class citizens alongside the GPT and human results. My understanding is that the behavior of these methods is much less mysterious than that of GPT or humans -- if the background knowledge given to the inductive programming methods were perfectly aligned with P3, then we would see perfect agreement in the results. So really the results for these methods are just a function of the background knowledge. I’m not sure what we are meant to take away from these results in particular.” ---
> We agree with the comment, since the behavior shown by these inductive programming methods is more expectable. Our initial intention was to add MH since we wanted to study different learning paradigms and compare their bias with respect to humans. In a second version of the paper we added Louise following the recommendation of an IJCAI reviewer. In the final version of the paper, we will put the main focus of the experimental evaluation on humans and language models, with IP models more in the background. This will also allow for more space for the analysis of language models and the better narrative.
>
> --- R: “The link in footnote 1 (https://bit.ly/Code3357) cannot be accessed (permission error)” ---
> Now it seems to be working. We do not know what may have happened.
>
> --- R: “For both humans and GPT, I would be curious to see if the results are robust to changes in the syntax. For example, instead of 0 and 1, what about a and b, or apple and banana, # and %, etc.” ---
> We tried with many variations of prompts and several changes of alphabet. Actually, we already did some of these experiments, but we did not include them into the supplementary material. For instance, we did experiments where 0 and 1 were replaced by a and b. The (anonymised) code can be found here:
> https://colab.research.google.com/drive/1uP4y-Rw-np1OCu8jnjKjtd7vlnn5YTi-?usp=sharing#scrollTo=hadAupflYP7W
> as well as a summary of these results (for four recursive concepts) also using some other prompts:
> https://docs.google.com/document/d/1WHmVg3pUbbk4wQ2QUxcDNzkSlB7UznA2cwGB1-7WgzU/edit
> Similarly, we also did some experiments with the terms ‘Apple’ and ‘Banana’ (and ‘Chair’ for the empty string). They are found here:
> https://colab.research.google.com/drive/1xSW372sFhvu4xD6RI6KPA4B19U7OJOo5?usp=sharing#scrollTo=ynFjqgOLnuot
> If we compare with Figures 1 and 2 in the appendix, we see that the results are not very different.
> We can complete these experiments with more repetitions and add them to the supplementary material.
>
> --- R: “When presenting the non-human approaches with WS, then WS + AS I, then WS + AS I + AS II, are the methods “reset” after each phase? For example, with GPT-3, does it see the full history of phases, or is just given each phase independently? I am guessing the latter. If so, is it possible that the humans have an inherent advantage over the machines? Intuitively, from the sequence, the humans could infer the relative complexity of the examples in each set, giving some additional hints about the target concept.” ---
> The experiment is done with the methodology as guessed by the reviewer (each phase independently). For humans this is not possible to do, but we don’t think this is an advantage, as the instances are not ordered by complexity.

---

> > ### Comment · Reviewer_HZ79 · 2021-08-14
> > **Thank you for your response**
> >
> > Thanks to the authors for addressing my comments and questions. After reading the response and the other reviews, I have decided to adjust my score to 6. I still believe that the paper's positives outweigh its negatives and I would be happy to see it accepted, but I recognize the concerns raised by other reviews.
> >
> > In particular, Reviewer yfFz raises important questions about the relevance of machine teaching. The experiment suggested by the reviewer, to use randomly selected examples instead of elements of the witness set, seems important towards justifying the machine teaching framing.
> >
> > I also remain concerned about the reliance on such a small set of concepts (15 is still too small). The limited access to the GPT API is unfortunate, but we ultimately have to make a judgement about whether the paper in its current form contains enough empirical evidence to justify the claims.
> >
> > Thanks again for addressing my other comments and questions.

---

> > > ### Author Response · Authors · 2021-08-15
> > > **Relevance of machine teaching and economy of the experimental design**
> > >
> > > We understand that judgements have to be done with existing evidence. In the P3 teaching book, we have the following number of non-redundant concepts for each program length:
> > >
> > > 1 -> 1
> > > 2 -> 4
> > > 3 -> 12
> > > 4 -> 35
> > > 5 -> 97
> > > 6 -> 269
> > > 7 -> 577
> > > 8 -> >1000
> > >
> > > When we sampled one concept per program length (stratified distribution), the trends were relatively consistent already with all systems, especially at the lower end, and we decided to extend the experiments on the more complex end (section 7 in the appendix) and concepts of algorithmic character (section 6 in the appendix). So we really increased the number of concepts where we thought it was most necessary, also following the suggestions of the IJCAI reviewers. We’re nonetheless working on the experiments with further concepts, which we will report by linking them to the code repo (still anonymously), but we don’t expect things to change significantly.
> > >
> > > In one of the recent responses to reviewer yfFz we have clarified why machine teaching is actually fundamental for the objective of the paper, and one of the major methodological contributions. The minimal teaching-size witness sets are precisely those that can collect most information per experiment. Using witness sets with many examples wouldn’t prove much in most cases, as many language biases (and learners) make the same choices when there is plenty of evidence (as they can discard most concepts as they become incompatible, not because of the learning bias). Machine teaching is the experiment-efficient approach to evaluate the questions we address in this paper. We didn’t think that doing a less efficient experimental design was appropriate, especially as experiments have associated costs (human time and compute). Using WS + AS I + AS II in sequence always explores “sufficient” sets for P3. Using AS I + AS II or any other sequences that do not build on sufficient sets would allow for some other identifications for P3: either we should change the concept that really emanates from the set or we would be testing the wrong concept, as we explain in more detail in the response to yfFz. Somehow reluctantly, we have agreed with reviewer yfFz to do the experiments with random sets, but they will simply introduce errors of different kinds.
> > >
> > > We also think that we are giving the right message in the paper: if some other researchers would like to evaluate the existence of a simplicity prior relative to another language (different from P3) and with other learners (e.g., future generations of language models), the procedure requires to build a teaching book for the language, choose concepts carefully with a stratified distribution on the range of complexities, and use the minimal witness sets to get maximum information per concept.
> > >
> > > We regret the update in the score, but we really thank the reviewer for their positive view of the paper overall.

---

### Official Review · Reviewer_s6LA · 2021-07-12

**Rating:** 6
**Confidence:** 3

**Summary:**

This paper studies the type of patterns learned by large pre-trained language models (LMs) in a few-show setting. The authors differentiate between external patterns (e.g., common-sense or world-knowledge), and internal patterns, such as ABAB, and focus on the latter. They frame this as a teaching problem with strong priors, and study the size of the example set (number of shots in the few-shot setting) compared to the length of explanation that can be provided to the model in order to teach it the given pattern. The authors focus on the simplicity prior, and study whether it is implicitly encoded in such large models.
Experiment that compare LMs, program induction systems and humans are performed. The main findings are that LMs perform as well as humans in few-shot settings, but the two populations differ in their ability to generate explanations.

I would say upfront that I am not an expert in this topic, and have found many points in the paper unclear, perhaps due to this. However, I wasn't fully convinced by the setup used by the authors. I am looking forward to the author response that might clarify some of these points.


**Limitations And Societal Impact:**

Yes

**Main Review:**


The main concern I have with this paper is that I am not sure how the concept of witness set as used in this paper is learnable by a machine or a human. Looking at the examples in Table 1, there are numerous ways of generalizing the first WS, some of them of similar simplicity (e.g., copy the first character, copy the last character, copy the entire sequence, etc.). Moreover, it is impossible to learn C4 from the single WS example. In fact, none of the examples contain the first rule (if input is '', print '00'), so how can any system be expected to learn this?
This makes me unsure of the interpretation of the results in this paper.


Other questions and comments:

1. Can you provide a formal definition of a concept? The authors discuss it rather informally in the intro and related work (e.g., the concept of an odd number), but then treat it as a function in #150 and onwards.
2. Q1 asks "Is a language model such as GPT able to capture algorithmic patterns from very few examples selected with a simplicity prior?", but then Q2 asks "Does accuracy increase from the theoretically minimal witness sets needed for teaching the concept with additional examples?" From the definition of "capturing a concept" in #154 we have "iff for every S we have that p 􏰑 S iff c 􏰑 S.". In what sense does accuracy improve once you've captured a concept?
3. #193: in which sense can a language relate to "the way language models work internally (transformers)"?
4. Please add random/majority baselines to the figures.


Minor:
#41 story -> history?


**Time Spent Reviewing:**

4

---

> ### Author Response · Authors · 2021-08-09
> **Response to reviewer "s6LA"**
>
> --- R: “The main concern I have with this paper is that I am not sure how the concept of witness set as used in this paper is learnable by a machine or a human. Looking at the examples in Table 1, there are numerous ways of generalizing the first WS, some of them of similar simplicity (e.g., copy the first character, copy the last character, copy the entire sequence, etc.). Moreover, it is impossible to learn C4 from the single WS example. In fact, none of the examples contain the first rule (if input is '', print '00'), so how can any system be expected to learn this? This makes me unsure of the interpretation of the results in this paper.” ---
> The set of examples in a witness set is finite and necessarily incomplete to distinguish among all compatible concepts. The identification is only possible in these learning scenarios when there are strong inductive biases, as is the case whenever one-shot or few-shot learning takes place. This also happens here (with the simplicity bias explicitly employed in the machine teaching setting). The witness sets are as minimal as they can be on purpose. Sometimes, as pointed out by the reviewer with concept C4, they have fewer examples than one would expect, or maybe miss some examples that humans would consider critically informative. This is also why we compare with humans, and we also added sessions with the AS I and AS II sets, to really evaluate the degree of alignment (or misalignment) in representation between humans and language models. Despite this misalignment, our experiments show that language models can capture abstract concepts that are generated by algorithmic languages such as P3, completely unrelated to the way language models work internally (transformers) and different from any natural language. In some cases, these are identified from very minimal sets, which indicates that the priors are aligned, which is exactly what we want to analyse in this paper. Were we using large witness sets with far more information, we would get that any capable learner (with their priors aligned or not) would be able to identify the concepts. In other words, minimal sets are crucial to analyse whether some other competing patterns (of “similar simplicity”) are ruled out to single out the right program, which would be “impossible” without the prior. We will bring some of the discussion in section 9 in the supplementary material to the main paper, especially the fact that there can’t be a sufficient witness set for *all* learners, just a minimal set for *one* learner, which would work for other learners to the extent their priors are shared.
>
> --- R: “Can you provide a formal definition of a concept? The authors discuss it rather informally in the intro and related work (e.g., the concept of an odd number), but then treat it as a function in #150 and onwards.” ---
> The introduction and related work covers a more general or more specific notion of concept, such as the set of all computable Boolean functions. In our case, concepts are P3 programs that receive a binary string as an input and generate a binary string as an output. In Table 2 of the supplementary material we include the concepts used in the experiment in P3, and, for the sake of comprehensibility, we show the concepts expressed as decision rules in Python. We will move some examples from supplementary to the main paper, and make the transition clearer from the introduction.
>
> --- R: “Q1 asks "Is a language model such as GPT able to capture algorithmic patterns from very few examples selected with a simplicity prior?", but then Q2 asks "Does accuracy increase from the theoretically minimal witness sets needed for teaching the concept with additional examples?" From the definition of "capturing a concept" in #154 we have "iff for every S we have that p is consistent with S iff c is consistent with S.". In what sense does accuracy improve once you've captured a concept?” ---
> In the theoretical machine teaching setting, with humans and even with inductive programming systems, if the concept has been captured, more consistent examples will not make it change. However, for language models we do not have such a monotonicity. In any case, Q1 and Q2 are expressed “on average”, and accuracy increases for humans and MH when aggregating for several problems, as we do in some of the tables and results. We will add the term “on average” for clarification.
>
> --- R: “#193: in which sense can a language relate to "the way language models work internally (transformers)"?” ---
> That sentence needs to be clarified. For a system such as MH, one can argue that Haskell is related to P3 or even to human language. However, language models are based on transformers that do not have any symbolic representations. After having been trained over massive corpora of natural language texts, it can distil human language, but P3 is a Turing-complete programming language not specialised for any domain and not common (if at all existent) in the training data.
>
> --- R: “Please add random/majority baselines to the figures.” ---
> Thanks for the suggestion, we will do it in the final version. In the case of the random baseline we need to consider some non-uniform distribution, because the number of strings is a countably infinite, enumerable set and we have to give lower probabilities to longer outputs.

---

> > ### Comment · Reviewer_s6LA · 2021-08-10
> > **Thank you for the clarificaitons**
> >
> > I think I have a better understanding of the paper now. A few followup questions:
> >
> > 1. I understand that the concepts are ambiguous, and require some prior in order to learn them correctly. I wonder what makes the *randomly* selected programs (#208) ones that require the simplicity prior in order to solve them. E.g., c1 can also be solved by taking the last character (at least when only considering WS and AS-I). Similarly, the simplest rule one could learn for each WS example is to simply return the label of the single example (e.g., for C1 and C2, return 0). I think much more discussion about the choice of the programs and the way that they require the simplicity prior is needed.
> > 2. Random baseline: agreed. I think the majority baseline is very important and could shed light on many of the results.
> > 3. In order to compare the priors of models and humans, wouldn't it make sense to measure the agreement between the predictions of the two?
> > 4. Minor: in Tab1, the description of c4 should have 011 and not 001.

---

> > > ### Author Response · Authors · 2021-08-10
> > > **Followup.**
> > >
> > > Thank you for reading our response and for posting new comments. Regarding the new followup questions:
> > > 1. The concepts are all taken from the teaching book (line #209), which means that they do not have any shorter program in P3, so a simplicity prior for P3 (what the “ideal” learner in P3 does) would be sufficient to identify them. But, because of the difference in description language L that humans and other systems have versus P3, we see that other concepts (equally simple or even simpler when expressed in their L) could be chosen by humans and other learners. The remarkable finding is that for languages that are so different, the invariance theorem still applies, and there is some degree of success in the extrapolations, which increases with larger example sets (where the differences in description language should weigh less) and with larger models (especially for larger concepts). We will explain the choice of programs and the language difference more plainly, leaving the details of the “teaching book” and the discussion of language alignments under the theory of machine teaching in section 3 for the supplementary material.
> > > 2. Ok.
> > > 3. We think this is what we do in Figure 4 (with accompanying text #317-324) but focussing on intrinsic extrapolations only, i.e., those that are not bound to human culture or even natural language (all our concepts emanate from an intentionally artificial P3). Comparing how language models and humans differ in terms of language syntax, semantics, culture and factual knowledge has been partly examined in other papers with a very different methodology (e.g., through perplexity measures).
> > > 4. This will be corrected. Thank you!

---

> > > > ### Comment · Reviewer_s6LA · 2021-08-11
> > > > **Thank you for the further clarification**
> > > >
> > > > I think I understand the main contribution of this paper at this point. I will update my score.
> > > >
> > > > As to Figure 4, I must have missed it. I think it would be interesting to break down these matrices per group or even per concept, as it seems that humans and machines are reasonably well aligned, though it is not clear if they are well aligned globally, or perfectly aligned for some concepts and not aligned at all for others.

---

### Official Review · Reviewer_yfFz · 2021-07-15

**Rating:** 6
**Confidence:** 3

**Summary:**

This paper proposes to empirically investigate to what extent large language models like GPT-3 learn (or "distill") a simplicity prior, i.e., a prior that favors concepts with shorter description length. The authors investigate this by randomly choosing 8 concepts/programs in the P3 language whose total number of P3 instructions number from 1 to 8, respectively. Using the framework of Telle et al. (2019), the authors are able to determine the "witness set," the minimal number of examples necessary for a learner with the correct simplicity prior to distinguish the intended concept from others. This witness set is fed to a learner, which is then evaluated on 5 randomly selected test examples of the concept. The authors also consider feeding additional sets of 2 and then 3 examples to the learner, each with its own random test set. The authors evaluate the accuracy of GPT learners, human learners, and 2 inductive program learners on these 3 test sets. They also elicit explanations from the learners.

**Limitations And Societal Impact:**

Yes

**Main Review:**

This is a very interesting paper, which has undertaken the difficult work of comparing GPT with both human learners and inductive programming learners in a systematic way. Furthermore, the paper contains some interesting findings, namely, that the advantage of larger GPT models is most pronounced when it comes to smaller, simple concepts, and that GPT models largely outperform humans on the tasks considered.

In terms of presentation, I found the paper difficult to understand at points. Even after reading section 3 it was unclear to me how witness sets could be so small and what exactly the authors meant by priors being aligned between learner and teacher, and I was only able to understand what the authors intended by reading the Telle et al. paper. In particular, I think what I was missing was that witness sets only need to distinguish the shortest satisfying program from earlier, shorter ones. Since much of section 3 is not used subsequently anyway, I would suggest that the authors instead use the space to emphasize this and how this leads to the experimental setup. I would also recommend the authors put their GPT prompts and the way they attempted to elicit explanations from GPT in the main text, since it's very hard to interpret any of the results otherwise.

Methodologically, I'm a little concerned that the experiments don't fully address what the authors are after. In particular, it seems like what we'd really like to know is what concepts the learner has learned, but instead we measure accuracy. This seems potentially problematic in situations where programs/concepts can have very similar outputs but differ drastically in their complexity, which is presumably the case here. (For instance, do we have any way of telling from the experiments that GPT hasn't learned the exact desired program except on a list of 10,000 exception inputs where it behaves differently?) Furthermore, even if we were satisfied with just accuracy, why not get the accuracy on a much larger sample size than 15 per concept (at least for the non-human learners)? I think if we were able to get reliable explanations from the models we could resolve these questions to some extent, but as the authors note this is difficult to do.

I also wonder whether the machine teaching dimension of the paper changes the story/experiments much. For instance, if instead of feeding the learners the optimal witness sets the learners just received random examples, would the plots change? (This experiment might be worth doing!) If not, these experiments may tell us more about the priors/biases of these learners in general rather than about how these learners perform when being taught, with ramifications for questions Q1 and Q2 in section 4.

Update after new experimental results: thanks for the new results; very interesting (especially for the VL experiments)! I increased my score.

**Time Spent Reviewing:**

5

---

> ### Author Response · Authors · 2021-08-09
> **Response to reviewer "yfFz"**
>
> --- R: “Since much of section 3 is not used subsequently anyway, I would suggest that the authors instead use the space to emphasize this and how this leads to the experimental setup. I would also recommend the authors put their GPT prompts and the way they attempted to elicit explanations from GPT in the main text, since it's very hard to interpret any of the results otherwise.” ---
> Thank you for your suggestions. We chose machine teaching to derive a well-founded controlled experiment to analyse the alignment of priors (learning bias) between the source of the concepts (what we call the teacher) and the learner. The learner is governed by a simplicity prior,  which is what we study in this paper; as the reviewer correctly points out, the teacher only needs to think of the shortest witness set such that all other simpler programs are inconsistent. This is the key idea that we will emphasise in this section, using more examples, appropriately swapping material from main text and supplementary material, keeping the paper inside the limits. We are confident that with these changes and further rewriting (especially for section 3) we make the paper easier to read, and doesn’t obscure the interesting findings, or limits the potential evaluation scores for our paper in terms of originality and contributions.
>
> --- R: “Methodologically, I'm a little concerned that the experiments don't fully address what the authors are after. In particular, it seems like what we'd really like to know is what concepts the learner has learned, but instead we measure accuracy. This seems potentially problematic in situations where programs/concepts can have very similar outputs but differ drastically in their complexity, which is presumably the case here. (For instance, do we have any way of telling from the experiments that GPT hasn't learned the exact desired program except on a list of 10,000 exception inputs where it behaves differently?) Furthermore, even if we were satisfied with just accuracy, why not get the accuracy on a much larger sample size than 15 per concept (at least for the non-human learners)? I think if we were able to get reliable explanations from the models we could resolve these questions to some extent, but as the authors note this is difficult to do.” ---
> We use accuracy since it allows us to estimate empirically the extent to which the learners have been able to induce the target concept correctly. For language models, as the reviewer says, there is no way to be sure, even if the explanations were correct, because the explanations are not the “programs” that are used by the language models to make the inferences, unlike the IP systems. In other words, the explanation might be wrong and the inferences correct, and the other way round. That’s one of the key motivations behind our methodology, which perhaps we didn’t emphasise sufficiently (but we will do in the next version of the paper). So for language models there is no way “to know what concepts the learner has learned” other than testing it. We considered the use of a sample size above 15 test examples per concept, but as we had humans in the comparison we had to limit this number to keep the questionnaires of reasonable duration. As the reviewer suggests we could increase the test examples just for the language models, but this may introduce some bias in the comparison, since using too many examples per concept would probably force us to include a higher number of larger (more informative) examples, making the distribution different from the one used with humans. We had to find a trade-off between the robustness of our results and the feasibility of a human study, which is key in our analysis. Nevertheless, we will do this extended analysis when comparing language models and other (non-human) systems, as suggested by the reviewer. We will also move some of the rationales behind our methodology from section 9 in the supplementary material to the main text.
>
> --- R: “I also wonder whether the machine teaching dimension of the paper changes the story/experiments much. For instance, if instead of feeding the learners the optimal witness sets the learners just received random examples, would the plots change? (This experiment might be worth doing!) If not, these experiments may tell us more about the priors/biases of these learners in general rather than about how these learners perform when being taught, with ramifications for questions Q1 and Q2 in section 4.” ---
> The use of the teaching size (not the teaching dimension) is the right approach for language models because it accounts for the amount of information we provide in the prompts. Three very short examples could provide less information than one long example. On the one hand, in the generation of concepts and examples we assume the learner uses a simplicity prior. On the other hand, the quantification of the information that a witness provides allows us to determine the alignment between the priors in the teaching setting and the results observed with the real learners (language models, IP systems or humans). While we think that the machine teaching size perspective is the right approach for the questions we make in the paper, we also use random selection for generating the extended batch examples (AS I and AS II) to be added to the witness set. However, by “random” we still have to set a distribution, and we still consider the size of the examples for the sampling. A uniform distribution doesn’t work for a countably infinite, enumerable set. But as the reviewer says, the experiment with other distributions is worth doing, and we will extend and complete some of the experiments with additional concepts that are already found in sections 6 and 7 in the supplementary material.

---

> > ### Comment · Reviewer_yfFz · 2021-08-13
> > **Thanks for your response!**
> >
> > Re: the second item above, I don't understand the point about how using more test examples would introduce bias in the comparison. Even if the test examples ended up being larger, they'd presumably be fed in (individually) after the same witness set prompt (as in Table 9). And even if this did hurt the comparison with humans, it still seems important to ascertain to what extent the GPT models truly learn the concept.
> >
> > Re: the third item above: sorry, I should have said "the machine teaching *aspect* of the paper" instead of "dimension"; I didn't mean to invoke the technical sense of "teaching dimension" :) But I think the point I'm trying to make is that since we don't have an experimental setting where the learners do *not* receive their first examples from the witness set, we can't tell if receiving the witness set first actually matters. And if it doesn't matter, then these results are not so much about what happens when we teach these models (by selecting witness sets, etc) as they are about what these models do with a small number of examples.

---

> > > ### Author Response · Authors · 2021-08-15
> > > **Testing and learning examples**
> > >
> > > Thank you for reading our response and asking for clarifications about the remaining concerns:
> > >
> > > Second item: By using larger examples in the *test* set, we may get different results just because the outputs are larger and the corrections of guessing them by chance should be different (mapping 0000 into 1010 by chance is less likely than mapping 0 into 1 by chance). This is something that happens in structured prediction, unlike classification or regression. But the reviewer has given us the idea of using a larger test set where the input strings may be larger, but the output strings follow the *size distribution* of the original sets we used (e.g., examples such as mapping 0000 into 1). With this constraint in the selection of test examples we can still generate hundreds of new test examples easily and get better estimates of the extent the model has captured the concept that can be compared with the results of humans and other systems. This is easy to do if the reviewer agrees with this approach.
> > >
> > > Third item: We can try with AS I and AS II (independently or together) or other different few shot sets without the witness set, at least for the language models. However, the witness set is more informative about the simplicity prior than choosing examples with some other distributions. More precisely, if the witness sets are not minimal, we can have two situations: (1) If they are sufficient but not minimal (for instance, they have more examples than strictly needed) then we measure the capability of the systems but less so their alignment with simplicity (this is what we explore by adding AS I and AS II after WS). (2) If they have insufficient examples to identify the concept even for P3, then it is quite unlikely that they would be sufficient for other languages, and the utility of these sets would be limited to explore simplicity. In addition, if they are insufficient, this means that for P3 they would really correspond to a *different* concept using the very simplicity prior of P3. Either we change the concept (and then we would be back to case 1) or we do the experiment with a “wrong” concept according to P3 simplicity. So using AS I and AS II not incrementally over WS would put us in case (2) and this should give higher errors (and of different kinds). In the end, this wouldn’t analyse simplicity very well, and is not a good alternative option (not even just less efficient) to the experimental setting we use in the paper. But we can at least corroborate that we get higher errors experimentally if the reviewer considers this is important.
> > >
> > > As the above two issues are solvable with some further experiments (which are straightforward to do), we would really appreciate it if the reviewer can consider this to update their score. Thank you!

---

> > > > ### Comment · Reviewer_yfFz · 2021-08-17
> > > > **Thanks for your (most recent) response!**
> > > >
> > > > Re: second item: thanks, I see. Yes, I think if it turns out that longer outputs are correlated with more errors it's fine to limit to shorter outputs as you suggest.
> > > >
> > > > Re: third item: yes, my concern is about case (2). Namely, I don't think there's been any empirical demonstration that insufficiency of the examples makes a difference to your reported accuracy metrics, which seems crucial to being able to draw the inferences you want to draw. So yes, I think experiments demonstrating that not feeding in the witness set changes the performance would be great. And I agree this is all solvable with further experiments, but I think the point is that the actual outcome of these experiments matters quite a lot here.

---

### Official Review · Reviewer_1G3n · 2021-07-16

**Rating:** 6
**Confidence:** 3

**Summary:**

In this paper, the authors compare the few-shot learning abilities of humans and GPT* language models (and inductive programming systems) from a machine teaching point of view. Models (and humans) are given a minimal witness set (examples using as few bits of information as possible) that should be sufficient to uniquely determine a program (function from a sequence of bits to another) under a pretty strict simplicity prior. The authors also examine whether additional examples (on top of the minimal witness set) help performance. They consider 8 different P3 programs of increasing complexity based on the number of operators. Examples are presented as English prompts.

The minimal witness set is often not sufficient for humans and LMs to generalize correctly. Nevertheless, there is some evidence that GPT models favor simplicity to some degree. Overall, larger GPT models generally perform better than smaller ones, but the correlation is not perfect. GPT-3 models often generalize better than the human candidates based on the few provided examples. However, humans sometimes provide acceptable explanations (for the shorter programs), but the authors could not get GPT models to explain their reasoning.


**Limitations And Societal Impact:**

Mostly yes. Limitations and societal impacts are discussed, albeit briefly, in the introduction and at the end of the paper.

**Main Review:**

The paper mostly explores LM few-shot learning [1] under the machine teaching setup in [2].

Experiments appear to be designed carefully. Multiple models are compared, both in terms of program complexity and number of examples given. The number of programs is low (only 8), which the authors justify as balancing between obtaining robust results and limiting the human effort needed. I would suggest keeping the current setup for the human comparisons, but greatly expanding the number of programs for the automated systems (much more than in the supplementary material).

Most of the paper is pretty clear, although I had trouble understanding section 3. In addition to the mathematical definitions, I would appreciate if the authors could explain more thoroughly in plain text.

The findings appear reasonably significant for researchers involved in language modelling and machine teaching. As language models are commonly used, understanding their behavior, limitations and how they compare to humans seems important.

Other remarks and questions for the authors:

L83: Is it really a failure?

Figure 1 is arguably overcrowded. It includes changes to both temperature and model size. For additional clarity, I think these should be presented separately.

In Figure 1b, what is the data condition (WS, WS +AS I, WS + AS I + AS II)?

You group 2 programs under each complexity category. Do you think that the paper would be more informative if the results of each program were presented on their own, or would it make the paper harder to read?

Does the order of examples matter?

The matrices in Figure 4 are symmetric (along the bottom-left to top-right diagonal), so most information is duplicated.

Why is the size of the additional examples not expressed in bits, as for the minimal witness sets?

Even with additional examples over the minimum witness set, performance of all systems and humans is still far from perfect. I would be interested in knowing how many examples (or bits) are needed to reach certain performance thresholds (if they are at all reachable).

If I understood correctly, the minimal witness set should allow to uniquely identify a program, under the assumption that shorter programs are preferred. On line 160, you mention "In case of equal l, ties are broken lexicographically." Does that mean that if a witness set is compatible with 2 programs of the same length, one program would be chosen arbitrarily? If that is the case, I would argue that the witness set is inappropriate to explain the chosen concept/program. Does this phenomenon occur in practice?

[1] Brown et al. (2020). Language models are few-shot learners. NeurIPS.
[2] Telle et al. (2019). The teaching size: computable teachers and learners for universal languages. Machine Learning.

**Time Spent Reviewing:**

5

---

> ### Author Response · Authors · 2021-08-09
> **Response to reviewer "1G3n"**
>
> --- R: “Experiments appear to be designed carefully. Multiple models are compared, both in terms of program complexity and number of examples given. The number of programs is low (only 8), which the authors justify as balancing between obtaining robust results and limiting the human effort needed. I would suggest keeping the current setup for the human comparisons, but greatly expanding the number of programs for the automated systems (much more than in the supplementary material).” ---
> In the supplementary material we focused on the areas where we had more variability (see sup. section 5), to extend from the original 8 concepts. The overall results for these 15 concepts support the original findings, with the trends becoming more clear and less dependent on the original choice of 8 concepts. But as the reviewer suggests, we will add even more concepts to further confirm the findings.
>
> --- R: “Most of the paper is pretty clear, although I had trouble understanding section 3. In addition to the mathematical definitions, I would appreciate if the authors could explain more thoroughly in plain text.” ---
> We will rewrite section 3 to include only the essential definitions and accompany them with plain-text clarifications, moving some other details to the appendix. In the end, machine teaching allows us to determine the minimal witness set in terms of size for a given concept, assuming a simplicity prior on the side of the learner. We chose this setting as a well-founded controlled experiment to analyse the alignment of priors (learning bias) between the source of the concepts (what we call the teacher) and the learner, which is what we study in this paper.
>
> --- R: “L83: Is it really a failure?” ---
> Yes, because we want GPT-3 to understand the sentence, and the only valid continuation for that sentence is Mary.
>
> --- R: “Figure 1 is arguably overcrowded. It includes changes to both temperature and model size. For additional clarity, I think these should be presented separately.” ---
> Thanks for your suggestion, we will improve the figure in the following version.
>
> --- R: “In Figure 1b, what is the data condition (WS, WS +AS I, WS + AS I + AS II)?” ---
> We say that it is “the average accuracies”, but we will be more explicit that Figure 1b includes all three batches (WS, WS +AS I, WS + AS I + AS II) and calculates a mean per complexity. This is dual to Figure 1a, where all the complexities are averaged and the results are disaggregated by batches.
>
> --- R: “You group 2 programs under each complexity category. Do you think that the paper would be more informative if the results of each program were presented on their own, or would it make the paper harder to read?” ---
> Originally we included the results for each program in the paper, but for the sake of comprehensibility, we decided to group them by complexity, and leave the results for each program in the supplementary material (sup. Section 5). As requested by the reviewers we will include more concepts for the next version. This means that finding the space for the detailed results in the paper rather than the supplementary material will become more challenging, but we will see what we can do when we restructure section 3.”
>
> --- R: “Does the order of examples matter?” ---
> In the case of language models, it does matter (see e.g., https://arxiv.org/abs/2104.08786). For that reason we show the average of ten executions. For the rest of the systems (and humans), the order is not so relevant.”
>
> --- R: “The matrices in Figure 4 are symmetric (along the bottom-left to top-right diagonal), so most information is duplicated.” ---
> Yes, we think that it helps reading one method against the rest. Eliminating half the matrix would not give us extra space, so we decided to keep it full.
>
> --- R: “Why is the size of the additional examples not expressed in bits, as for the minimal witness sets?” ---
> This is a good observation. Our machine teaching model is not prepared to consider extra examples (it could be extended but we preferred to use the original model), so AS I and AS II are generated in a different way, not considering size as is the case for WS.
>
> --- R: “Even with additional examples over the minimum witness set, performance of all systems and humans is still far from perfect. I would be interested in knowing how many examples (or bits) are needed to reach certain performance thresholds (if they are at all reachable).” ---
> We see some saturation for AS II over AS I in Figure 1 and 2 but the curves could still grow slightly. This can be explored in the next versions of the paper.
>
> --- R: “If I understood correctly, the minimal witness set should allow to uniquely identify a program, under the assumption that shorter programs are preferred. On line 160, you mention "In case of equal l, ties are broken lexicographically." Does that mean that if a witness set is compatible with 2 programs of the same length, one program would be chosen arbitrarily? If that is the case, I would argue that the witness set is inappropriate to explain the chosen concept/program. Does this phenomenon occur in practice?” ---
> Yes, we used that simplicity criterion to generate witness sets from concepts. We decided to keep the first program when there are ties in size, as a small percentage of these ties happen with programs that are different. Most of these ties happen when there are two programs of the same size that are equivalent. However, whether two programs are equivalent for every input is not easy to be determined in many of the cases we inspected (and not decidable in general for a Turing-complete language such as P3). Actually, by making the witness sets larger, these equivalent programs would never be distinguished. We can give more details of this phenomenon in the supplementary material.

---

### Official Review · Reviewer_HcAt · 2021-07-19

**Rating:** 6
**Confidence:** 3

**Summary:**

This paper attempts to ask whether language models have a bias that allows them to learn from fewer examples in cases where a rule is simpler. They compare exploitation of rule simplicity in a variety of models, including human participants. This prior towards simplicity is most strongly observed in larger models, but the minimal example set is rarely enough to train any system, so with the exploitation of simplicity is not a perfect prior.

The paper also provides an analysis of the capability of these systems to provide explanations of the rule, but only humans and the algorithm that they specifically select for this purpose were capable of providing such explanations.

**Ethical Concerns:**

This paper seems to have followed reasonable procedures in their research with human subjects.

**Limitations And Societal Impact:**

Because this paper has such a broad prior it is investigating, the results may be misleading if the true priors are more constrained than "simplicity". Otherwise, this seems to have a limited impact outside of understanding of the inductive bias of neural networks.

**Main Review:**

My main complaint with this paper is that it describes what's happening as a simplicity prior, or Occam's razor, but a more in-depth analysis of the pattern they observe might reveal that these systems have a prior towards specific types of rules or methods of compression, such as those that exploit hierarchical patterns or memorized chunks. Because there was a lack of analysis of the rule type, focusing only on an abstract idea of simplicity based on a particular method of compression, I found this unconvincing as a study of a general simplicity prior.

I was most impressed by the fact that there was a human study. This allows us to compare the priors of an artificial language model with the ability of a human to learn rules. I would like to see more papers like this, in that respect.

Minor issues / clarity:
- I wasn't satisfied with the attempt to explain what a concept was. Within the paper, this is left very vague.
- What does it mean to say that an explanation is "minimally acceptable"?
- "A more systematic analysis is needed to determine the optimal prompts that make the model capture a range of different patterns." This sentence seems to indicate that this will be the focus of the paper, but it isn't my understanding of the purpose of the paper.
- "most representative AI systems" - This is a really unclear claim
-  I found the example about "go stop go" pretty confusing. I wonder if you could come up with a better example that doesn't involve iffy phrases like "go stop go stop" or even obscure English idioms like "ready steady go".
- I was confused as to the purpose of the paragraph on liens 114-123. It didn't seem to particularly relate to the rest of the paper.
- "GPT-3D-T0 is one of the most similar systems to humans" -  What does this mean?

 Grammar:
- line 163 "being w"

**Time Spent Reviewing:**

3

---

> ### Author Response · Authors · 2021-08-09
> **Response to reviewer "HcAt"**
>
> --- R: “My main complaint with this paper is that it describes what's happening as a simplicity prior, or Occam's razor, but a more in-depth analysis of the pattern they observe might reveal that these systems have a prior towards specific types of rules or methods of compression, such as those that exploit hierarchical patterns or memorized chunks. Because there was a lack of analysis of the rule type, focusing only on an abstract idea of simplicity based on a particular method of compression, I found this unconvincing as a study of a general simplicity prior.” ---
> This is a very important point. The language we use and the diversity of concepts support the hypothesis of a general simplicity prior over a group of ad hoc patterns. But even if the language models are retrieving particular methods of compression, this favours simplicity, if the model chooses one pattern that has a shorter representation over other possible patterns that have a larger representation. We’re not claiming that language models are universal compressors and could implement all possible algorithmic ways of compressing data. Indeed, in the supplementary material (section 6), we show what happens with concepts with loops, where we show that the performance of language models is poor. This contrasts with the good results when using AS II in hard and very hard concepts in the supplementary material (section 7). This is a more detailed analysis of rule type versus size, as suggested by the reviewer. Our claim, from the analysis of language model sizes, is that this is more related to the capability of the systems, with more abstract (algorithmic) rules possibly being featured in larger language models in the future.
>
> --- R: “I wasn't satisfied with the attempt to explain what a concept was. Within the paper, this is left very vague.” ---
> A concept is a P3 program that receives a binary string as an input and generates a binary string as an output. In Table 2 of the supplementary material we include the concepts used in the experiment with P3, and, for the sake of comprehensibility, we show the concepts expressed as decision rules in Python. We will move some examples to the main text.
>
> --- R: “What does it mean to say that an explanation is "minimally acceptable"?” ---
> The sentence is confusing; we wanted to express that only the explanations given by humans and MH are valid, in opposition to the majority of the explanations from GPT-3, which were incorrect or useless. We’ll rewrite it.
>
> --- R: “"A more systematic analysis is needed to determine the optimal prompts that make the model capture a range of different patterns." This sentence seems to indicate that this will be the focus of the paper, but it isn't my understanding of the purpose of the paper.” ---
> We wanted to express that this would be a goal for another paper. We will rephrase this. Thank you.
>
> --- R: “"most representative AI systems" - This is a really unclear claim” ---
> Yes, and unnecessary. We will replace this by “are a kind of AI systems”. Thank you.
>
> --- R: “I found the example about "go stop go" pretty confusing. I wonder if you could come up with a better example that doesn't involve iffy phrases like "go stop go stop" or even obscure English idioms like "ready steady go".” ---
> Thanks for your suggestion, in the new version we will employ a better example of a non-idiomatic pattern versus an idiom.
>
> --- R: “I was confused as to the purpose of the paragraph on liens 114-123. It didn't seem to particularly relate to the rest of the paper.” ---
> In this part, we wanted to present papers that show that language models are able to perform tasks that involve abstract generalisations and their comparisons with human performance in those tasks. We’ll clarify this.
>
> --- R: “"GPT-3D-T0 is one of the most similar systems to humans" - What does this mean?” ---
> The sentence will be rewritten to: “Considering Figure 4, GPT-3D-T0 shows a similar pattern of behaviour (correct/incorrect results) with respect to humans in comparison with the rest of analysed systems”.

---

> > ### Comment · Reviewer_HcAt · 2021-08-09
> > **Thank you for the author response**
> >
> > Unfortunately, I don't feel satisfied about your claim of general simplicity. Though it is qualified when discussing loops, it still remains the core "selling point"/summary, and it is not justified by the mixed findings in terms of the inductive biases. If you reframed the paper in terms of the more specific inductive biases of the model, rather than in terms of a general bias toward simplicity, I would feel more positively about your claims. As the paper stands, I'm keeping my score.

---

> > > ### Author Response · Authors · 2021-08-10
> > > **Understanding the concern about “general” simplicity**
> > >
> > > Thank you for reading the response and expressing the remaining concern. We need to understand what you mean by “general” simplicity, for any reframing or change to the paper to really resolve the concern. Q5 (“a stronger preference for simple concepts”) is one of the key questions we address in the paper, and the one that is highlighted in the title. In section 9 of the supplementary material (lines #154-158 in particular) we give a precise definition and full discussion of what we mean by a model having “distilled a simplicity prior”. The nuance of the results is mostly due to our concepts being generated in the “teaching book” such that no shorter program exists *in P3*, which is not the language used by humans and the other systems. But the positive results cannot be explained unless there’s some bias in the models for few-shot learning. Our definition does not distinguish whether this bias is based on a specific collection of patterns with short descriptions (while failing on other specific patterns) or is based on a more “general” (Turing-complete) simplicity bias. In both cases, shorter patterns are preferred, and we must conclude that Occam’s razor is distilled to the extent of the observed accuracy in the results. Would a clarification along these lines work? For instance, explaining in a prominent part of the paper that we do not necessarily mean a “general” (Turing-complete) simplicity bias but possibly a collection of specific patterns working as an inductive bias that gives preference to shorter concepts?

---

> > > > ### Comment · Reviewer_HcAt · 2021-08-13
> > > > **Clarification**
> > > >
> > > > I meant that describing this in your paper as a general simplicity prior is a misrepresentation of more constrained and complex results from your actual experiments. The current framing in the paper is really overclaiming.

---

> > > > > ### Author Response · Authors · 2021-08-15
> > > > > **We don’t claim a “general” simplicity prior being distilled *by the learners***
> > > > >
> > > > > Thank you for further clarifications about the perceived issue. We’re sorry that we used the word “general” in our first response, because this seems to have complicated our discussion.
> > > > >
> > > > > By "simplicity prior" we always mean the same, a language-based simplicity bias. Consider for instance a language L and any concept c in the language. If a learner is given some example set S compatible with c such that all other compatible concepts with S have a longer representation in L, and the learner chooses c, then the learner is following a simplicity prior with respect to L. As we choose the *minimal* set for c, we ensure that we have plenty of other consistent concepts in L and the prior plays a major role in this selection. This is very important, because if S were made very large, it would exclude many alternative concepts (of any size) that would become inconsistent with these extra examples, and the experiment would be less informative. That is what our experimental setting analyses.
> > > > >
> > > > > When we use universal languages such as P3 that allow for any possible concept, then we explore the phenomenon more broadly and without any constraint in representation. We consider all terminating P3 concepts ordered by program length, and we systematically sample the concepts in a stratified way, by this program length, trying to be as “general” as possible in our sample for P3 concepts. There is no other partial selection or preference other than program length. But this is still simplicity with respect to P3 and, of course, any learner not based on P3 may be partial in the kinds of concepts it can identify, or a simplicity prior being applied in a partial way. We actually study this for GPT-3.
> > > > >
> > > > > We chose P3 because it is sufficiently different from any internal representation language any of the learners analysed could use (including humans) so that the invariance theorem could play a role in extrapolating simplicity in P3 to simplicity in other languages. That's how far we go. We didn’t use the term “general simplicity prior” in the paper. We don't want to make any overstatement.
> > > > >
> > > > > In fact, we raise the issue of language misalignment in the paper several times, and we mention that the "prior (Occam's razor) depends on the representational language", and that "no system prior is perfectly aligned with the simplicity prior that derives from P3 program size. This is expected as the internal representations of these systems are different".
> > > > >
> > > > > We really want to resolve the issue, so if we are still misunderstanding what the issue is, please indicate the parts that we need to rewrite or reframe, and we will do the changes.

---

### Author Response · Authors · 2021-08-09
**General response to the reviewers and chairs**

We thank the reviewers for their generally positive comments, in terms of the novelty of the questions we present, the adequacy of the methodology (using machine teaching but also with particular appreciation of the study with humans) and the importance of the findings for the NeurIPS community.

Some comments relate to giving more relevance to the results from language models (over the IP systems, which were extended following the reviews from IJCAI). This change is easy to make and can help us focus more on the key questions of the paper. Some extra space for this can be obtained by restructuring section 3, as suggested by one reviewer.

Some reviewers indicated that we should have tried with more than 8 concepts, the ones that we used for the human questionnaire. The supplementary material already included the GPT-3 results for four extra algorithmic concepts (sup. section 6) and three additional complex concepts (sup. section 7), partly motivated by a request from the IJCAI reviewers. These extra concepts focused on those cases where the variability in the original results was higher (see sup. section 5). The overall results for the full set of 15 concepts (sup. table 13) support the original findings, with the trends becoming more clear and less dependent on the original choice of 8 concepts. Nevertheless, we will add further concepts and instances with GPT-3 (prioritising the informativeness of new concepts over just the number) to give extra additional support to those findings that do not require the comparison against humans. We have requested OpenAI to be granted the option to buy more quota for their API and we expect to have further results with GPT-3 in the appendices soon.

One reviewer suggested that we could do some more experiments with variations of the prompts and the alphabet of the concepts. Actually, we had already done experiments along those lines (see the links in the response to reviewer HZ79) but we didn’t include them in the supplementary material as the results were similar to the reported results. We will include them in the next version of the supplementary material.

In the specific comments to the reviewers we also clarify some misunderstandings, especially those related to the motivation behind the machine teaching size setting and the use of strictly minimal witness sets for the first batch. While some identifications from such small sets of examples look hard or even “impossible” (as pointed out by reviewer s6LA), these can only happen if there is a clear alignment of priors. Using witness sets with many examples would be uninformative for the main question we analyse in the paper: whether language models have a simplicity prior or not.

According to all this and the detailed responses we give in the specific comments, we would really appreciate that the reviewers consider readjusting their scores.

---

### Author Response · Authors · 2021-08-30
**Generating the first batch randomly**

We already have the results for the comparison between the approach when the first batch is generated using the teaching book (WS, as for the original experiments) and the approach when the first batch is generated randomly. This experiment was suggested by reviewer yfFz.

More specifically, we want to compare the effect of the first batch, as a witness set (WS) chosen using machine teaching, with an alternative procedure of choosing the first batch as a random set (let us call it RS 0). The most important thing for making this comparison fair is that both sets must have the same sizes. If we allow for larger sets for WS than RS 0, or vice versa, then this would be an unfair comparison, since one would have more “bits” than the other.

With this constraint, there are still many choices to make the experiment informative, and it is hard to find a setting for a fair comparison.

This is the configuration that we think best reflects the question from the reviewer. We kept the AS I, AS II and test set as the original experiments, so that we only focus on the effect of the first batch, keeping everything else equal. We also kept the size $s$ of the original WS for each concept. Since this $s$ could give advantage to the old method (as they would never have overlap with AS I and AS II), we regenerated both approaches as follows.
We generated the WS choosing from the witness sets of size $s$ in the teaching book, always checking that the examples are not in the test set.
We generated a RS 0 choosing examples randomly of sets of size $s$, always checking that the set is neither in the witness sets for that concept in the teaching book nor its examples are in the test set.
We also chose this configuration as otherwise for small complexities there would not be many instances to choose from, given a limit on the size. Still, in those cases where A or B couldn’t complete the set, we used the original WS.

Note that this configuration plays in favour of approach B, as we choose from a random distribution for Test and RS 0, while we choose from the teaching set distribution for WS. It’s hard to think of a totally fair configuration, so we decided to choose against A.

We generated 5 VL-C + 40 L-C + 40 H-C + 40 VH-C = 125 concepts, the same for both experiments.

Experiment A gives these results:

```
Complexity    Batch	Results
VL            WS       0.330000
VL            AS I     0.550000
VL            AS II    0.590000

L             WS       0.168750
L             AS I     0.341667
L             AS II	0.409167

H             WS       0.123504
H             AS I     0.146581
H             AS II	0.306838

VH            WS       0.048571
VH            AS I     0.186667
VH            AS II	0.300000
```

The results of experiment A are very similar to those reported the other day, as they both use a WS from the teaching book. Experiment B gives these results:

```
Complexity	Batch	Results
VL            RS 0     0.180000
VL            AS I     0.340000
VL            AS II	0.750000

L             RS 0     0.150833
L             AS I     0.245417
L             AS II	0.404583

H             RS 0     0.095299
H             AS I     0.179060
H             AS II	0.347009

VH            RS 0     0.175238
VH            AS I     0.182381
VH            AS II	0.313333
```

When comparing the results between WS (from experiment A) and RS 0 (from experiment B), we see that WS gives better results, especially for VL. They seem to produce more informative examples with the same size.

However, the results for VH in experiment B are higher than expected. In this VH case, the value of $s$ is usually large and experiment B has more to choose from. This means there is more coincidence between the outputs of AS 0 and the test set. In other words, the test set doesn’t try to cover a range of different cases as WS tries to do, but it is generated randomly with many repeated outputs that also benefit RS 0. Controlling for cases where the target output appears four or more times in the RS 0 gives a value of 0.067, which better fits the decreasing trend in experiment B. In any case, let’s remember that we choose from a random distribution for the test set, which is used to evaluate RS 0 (sampled from the very same distribution) and WS (sampled from a different distribution, the witness set). Also note that it’s not the goal of our paper to find the sample that gives the best results, but the one that best corresponds to the simplicity prior.

Overall, we think that these results are in agreement with the machine-teaching based WS being the most informative sets to represent the inductive bias used for P3, which translates partially to other representations, at least for concepts of small and medium complexity.

We’re working on extended experiments for the final version of the supplementary material (and more variants of all these experiments), but decided to report on the first one about a larger number of concepts last week and this one today because we think this information may be useful for the last stretch of the discussion and for refining your final scores. Thank you!

---

> ### Comment · Reviewer_yfFz · 2021-09-01
> **Thanks for the new results**
>
> I increased my score.

---

### Decision · Program_Chairs · 2021-09-27

**Decision:**

Accept (Poster)

**Comment:**

This paper seeks to provide more insight into the few-shot learning capabilities of language models by pitting them against humans on a set of algorithmic tasks under the framework of machine teaching. Specifically, the goal of the paper is to identify whether language models learn to prefer simple explanations for algorithmic rules. Reviewers appreciated the thorough experimental design, in particular the inclusion of a high-quality human study. There were some concerns raised about clarity, claims, and fit for NeurIPS, but I think these can be improved for the camera-ready version based on the author's rebuttal.